SPECIAL ISSUE
THE EXTRACELLULAR ENVIRONMENT

# Axonal defasciculation is restricted to specific branching points during regeneration of the lateral line nerve in zebrafish

Rohan S. Roy[1],* and A. J. Hudspeth[1,2]

## ABSTRACT

Peripheral nerve regeneration requires precise selection of the appropriate targets of innervation, often in an environment that differs from that during the developmental wiring of the neural circuit. Severed axons of the zebrafish posterior lateral line nerve have the capacity to reinnervate mechanosensory hair cells clustered in neuromast organs. Regeneration represents a balance between fasciculated regrowth of the axonal bundle and defasciculation of individual axons into the epidermis where neuromasts reside. The cues that guide pathfinding during regeneration of the posterior lateral line nerve are unknown. Here, we show that regenerating axons selectively defasciculate through distinct gaps in the epidermal boundary layer. We found that the gene *col18a1a*, which encodes the secreted heparan sulfate proteoglycan collagen XVIII, is expressed by the neuromast and by a subset of Schwann cells that are located at the points of axonal defasciculation. Furthermore, we observed aberrant axonal branching at inappropriate locations during nerve regeneration in *col18a1a* mutants. We propose a model in which collagen XVIII patterns the basement membrane to affect the precision of axonal navigation.

KEY WORDS: Hair cell, Collagen XVIII, Synapse, Axon guidance, Schwann cell, Extracellular matrix

## INTRODUCTION

Damage to sensory neurons of the peripheral nervous system is often accompanied by detachment from end organs and degeneration of axonal fragments distal to the site of injury (Waller, 1851). The ability of damaged peripheral axons to regenerate and reinnervate their original targets varies between vertebrates and depends on the site and type of injury (Deumens et al., 2010). In particular, regeneration of the peripheral neurites of spiral ganglion neurons in the mammalian cochlea is lacking. Retraction of afferent fibers from the sensory epithelium of the organ of Corti occurs either directly through insult to spiral ganglion neurons or is secondary to the death of the mechanosensitive hair cells that are presynaptic to afferent terminals (Liberman, 2017;

[1]Laboratory of Sensory Neuroscience, The Rockefeller University, New York, NY 10065, USA. [2]Howard Hughes Medical Institute, The Rockefeller University, New York, NY 10065, USA.

*Author for correspondence (rroy@rockefeller.edu)

R.S.R., 0000-0001-7770-8304; A.J.H., 0000-0002-0295-1323

Shibata et al., 2011). Regardless of the cause, the retraction of afferent endings is irreversible and results in permanent sensorineural hearing loss.

Compared to mammals, the zebrafish (*Danio rerio*) possesses a greater capacity for regeneration of both the central and peripheral nervous system (Rasmussen and Sagasti, 2017). The posterior lateral line (pLL) of the zebrafish has emerged as a model for studying the regeneration of peripheral sensory nerves (Ceci et al., 2014; Lozano-Ortega et al., 2018; Villegas et al., 2012; Xiao et al., 2015). The pLL comprises discrete neuromast organs that are deposited by a migratory group of primordial cells on the lateral surface of each side of the fish (Metcalfe, 1985). Each neuromast includes 10 to 20 sensory hair cells clustered in a rosette and surrounded by non-sensory supporting cells (Ghysen and Dambly-Chaudière, 2007). Neuromast hair cells, which are genetically, structurally and functionally similar to those of the inner ear, are polarized to detect the direction of external water flow and thus to mediate swimming behaviors such as rheotaxis, schooling and predator avoidance (Pickett and Raible, 2019).

Neuromasts are innervated by the pLL nerve, which contains predominantly afferent fibers postsynaptic to hair cells and a few efferent fibers presynaptic to hair cells (Haehnel et al., 2012; Haehnel-Taguchi et al., 2018; Manuel et al., 2021). On each side of a larva, the cell bodies of ~40 to 60 afferent neurons reside in a pLL ganglion (Haehnel et al., 2012; Raible and Kruse, 2000). These bipolar neurons have short central projections to the hindbrain and long peripheral axons that extend along the horizontal myoseptum as a nerve from which individual fibers defasciculate serially to innervate neuromasts (Fig. 1A). Although peripheral axons are initially towed by a primordium lateral to the epidermal basement membrane, axon shafts are repositioned medial to the basement membrane, whereas defasciculated arbors remain embedded in the neuromast (Raphael et al., 2010).

The larval pLL offers a simple and accessible neural circuit in which to study axonal pathfinding, target selection and hair cell reinnervation after nerve lesion. Regeneration of the larval nerve is quick: an axotomized nerve can reinnervate all neuromasts within 2-3 days post lesion (2-3 dpl) (Ceci et al., 2014; Villegas et al., 2012; Xiao et al., 2015). Concomitant with Wallerian degeneration of distal fragments, cut axons first bridge the lesion with the assistance of neighboring Schwann cells (Ceci et al., 2014). Following the retrograde clearance of axonal debris, the proximal stump regenerates along stripes of surviving Schwann cells: the bands of Büngner (Ceci et al., 2014; Xiao et al., 2015). Most cut axons reinnervate neuromasts distinct from those that they initially innervated during development, albeit with a preservation of global somatotopy (Ceci et al., 2014; Lozano-Ortega et al., 2018). This pattern suggests a competition between neuron-intrinsic and -extrinsic cues to the regenerating growth cone that affects neuromast selection. Only a few axonal growth cones defasciculate

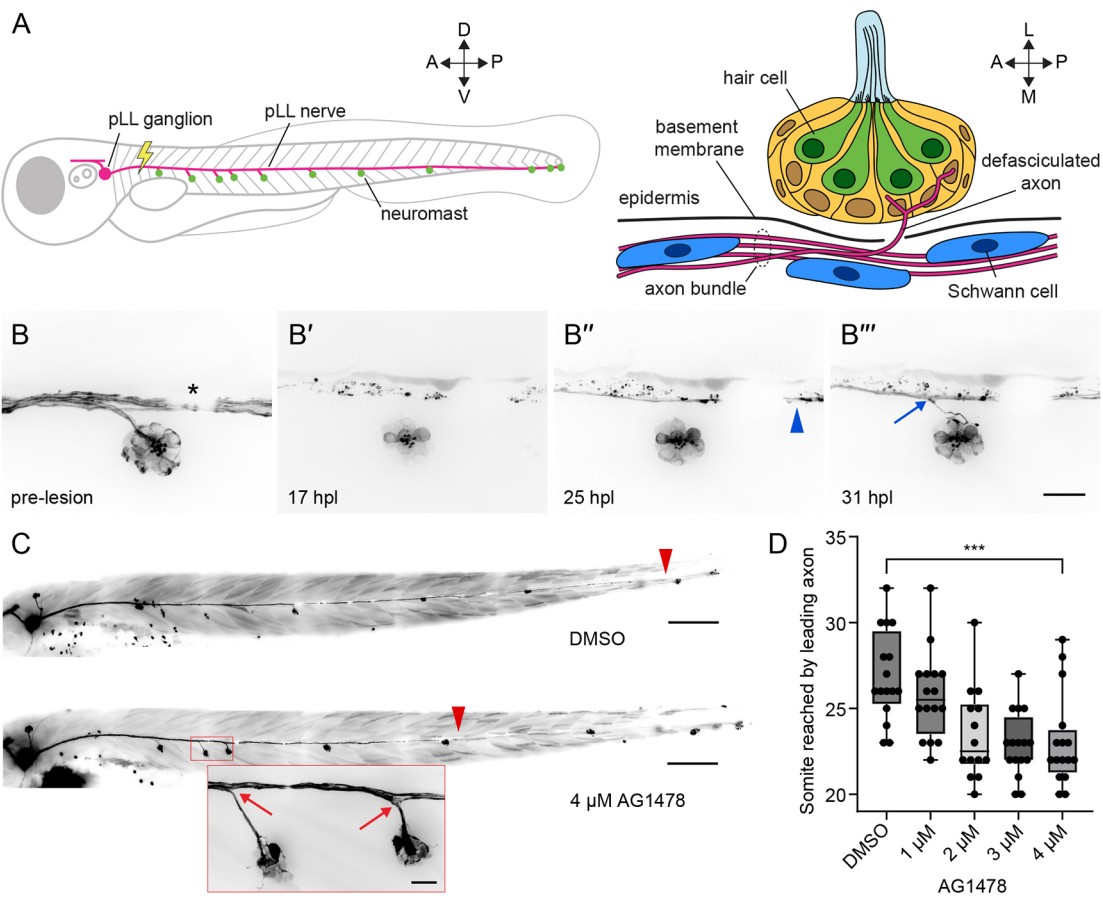

**Fig. 1. Regeneration of the pLL nerve occurs in stages.** (A) A diagram of the larval zebrafish pLL (left, parasagittal view) and an individual neuromast (right, coronal transverse view). For all experiments, the nerve was lesioned at the level of the fourth somite. (B-B‴) Individual frames from a timelapse video portray the re-innervation of a neuromast after nerve lesion in a 4 dpf *Tg(HGn39d; myo6b:actb1-EGFP)* larva. (B′,B″) Wallerian degeneration occurred within 24 h post lesion (24 hpl; B′) and was followed by passage of a leading axon (blue arrowhead) past the denervated hair cells of the neuromast (B″). (B‴) Six hours later, an individual follower axon defasciculated (blue arrow) from the axon bundle to enter the neuromast. The asterisk marks a melanocyte that blocked underlying fluorescence. (C) Representative images show 6 dpf *Tg(HGn39d)* larvae 2 days after nerve lesion and recovery in either 1% DMSO (top) or 4 µM AG1478 (bottom). Leader axons are marked with red arrowheads and defasciculation of follower axons is marked with red arrows (inset). (D) A box-and-whisker plot of the somite reached by the leading axon at 2 dpl and recovery in different concentrations of AG1478. Whiskers span minimal and maximal values; boxes indicate 25th to 75th percentiles; horizontal lines indicate median. ***$P<0.001$, unpaired $t$-test. $n=16$ zebrafish for DMSO, and 1 µM, 3 µM and 4 µM AG1478. $n=14$ zebrafish for 2 µM AG1478. A, anterior; P, posterior; D, dorsal; V, ventral; L, lateral; M, medial. Scale bars: 20 µm in B; 100 µm in C (inset 20 µm).

from the bundle at specific locations to enter the epidermis, pierce a basal layer of supporting cells and reinnervate hair cells. The factors that bias growth-cone decisions at each of these steps remain unknown.

The extracellular matrix (ECM) also plays an instructive role in the guidance of axons through mechanical and chemical signaling. Secreted heparan sulfate proteoglycans (HSPGs), glycoproteins with covalently linked polysaccharide heparan sulfate side chains, are embedded in the ECM and exhibit a variety of cell-signaling properties. HSPGs can signal directly through domains on the core protein and moieties on its sugar side chains, or indirectly through secondarily bound ligands (Poulain and Yost, 2015; Sarrazin et al., 2011). Collagen XVIII is one such secreted HSPG that has been observed in epithelial and vascular basement membranes throughout the body as well as in the peripheral nervous system (Halfter et al., 1998; Seppinen and Pihlajaniemi, 2011). In zebrafish, the gene *col18a1a*, which encodes the α1 chains of collagen XVIII, is needed for the ventral extension of motor growth cones from the spinal cord into the musculature and for the regeneration of severed retinal ganglion cell axons across the optic

chiasm (Harvey et al., 2024 preprint; Schneider and Granato, 2006). Collagen XVIII and its homologues have also been shown to affect axonal guidance in *Caenorhabditis elegans* (Ackley et al., 2001) and *Drosophila melanogaster* (Meyer and Moussian, 2009), as well as in the enteric nervous systems of the chick and mouse (Nagy et al., 2018). These observations suggest a conserved role in the patterning of the nervous system.

In the present study, we used a combination of timelapse imaging, bulk RNA sequencing and RNA fluorescence *in situ* hybridization (RNA-FISH) to characterize the defasciculation of individual axons from the regenerating axon bundle into the denervated neuromast. We find that *col18a1a* is expressed by the neuromast and a subset of Schwann cells that lie at the point of axonal defasciculation. Using a transgenic line that labels the epidermal boundary, we show that defasciculation occurs through specific gaps in the ECM. We additionally generated a *col18a1a* mutant that exhibits axonal branching at inappropriate locations during nerve regeneration. We propose a model in which collagen XVIII complexes with axon guidance cues in the extracellular matrix to refine and pattern axon fasciculation and defasciculation.

## RESULTS

### Regeneration of the pLL nerve occurs in stages

Peripheral afferent axons of the larval pLL nerve extend from somata in the pLL ganglion to the caudal end of the fish, innervating 7-11 neuromasts along the horizontal myoseptum. Individual axons defasciculate from the nerve to enter the epidermis and innervate hair cells located in the center of each neuromast (Fig. 1A). To label lateral line afferent axons and hair cell membranes for timelapse imaging, we used the *Tg(HGn39d; myo6b:actb1-EGFP)* transgenic line (Kindt et al., 2012; Nagayoshi et al., 2008). At 4 days post fertilization (dpf), we lesioned a 20 μm section of the pLL nerve between the ganglion and the first neuromast with a 355 nm laser. We then imaged the reinnervation of hair cells in the neuromast posterior to the cut site (Fig. 1B; Movie 1). As previously shown (Ceci et al., 2014), axon severing was followed by Wallerian degeneration of the axonal fragments distal to the cut site. As debris was being cleared in the 24 h after nerve lesion, a few pioneering axons extended past the denervated neuromast. About 6 h after the first pioneering axons bypassed the neuromast, a follower axon defasciculated ventrolaterally from the regenerating axon bundle to reinnervate the hair cells of the neuromast.

We perturbed regeneration by administering the ErbB tyrosine kinase inhibitor AG1478 after nerve lesion. Early administration of AG1478 prior to 2 dpf blocks Schwann cell migration along the peripheral axons of the pLL nerve through interference of ErbB-neuregulin signaling (Lush and Piotrowski, 2014; Lyons et al., 2005). Administration of AG1478 after Schwann cells have migrated onto the nerve does not affect the existing placement of Schwann cells but does hamper proliferation and renewal (Lush and Piotrowski, 2014). AG1478 administered after Schwann cells have ensheathed the nerve delays the traversal of pioneering axons across an axotomy cut site, but not the velocity of axon regrowth (Ceci et al., 2014). We asked whether blocking ErbB signaling at 4 dpf could also affect other stages of regeneration, especially the defasciculation of individual axons toward the neuromast. Recapitulating earlier results of Ceci et al., we observed a significant decrease in the distance along the trunk of the fish that the leader axon had reached by 2 dpl (Fig. 1C,D). Despite the delay to leader axon growth, we observed normal bundled growth along the horizontal myoseptum and defasciculation of follower axons from the bundle to reinnervate neuromasts (Fig. 1C). Although ErbB signaling might be involved in the initial passage of severed axons across the lesion site, it is dispensable for neuromast reinnervation. Our timelapse imaging indicated that nerve regeneration proceeds in two stages after axonal traversal of the lesion site: the extension of the axon bundle along the horizontal myoseptum in a leader-follower fashion and the defasciculation of follower axons from the bundle at specific locations along the trunk. The different locations and timing of these stages suggest that they are guided by distinct signaling cues.

### Neuromast hair cells transiently change gene expression after denervation

To identify guidance cues that may be coming from the targets of innervation, we performed bulk RNA sequencing on denervated hair cells. At either 1 dpl or 3 dpl, hair cells from *Tg(neurod: tdTomato; myo6b:actb1-EGFP)* transgenic larvae were isolated and sequenced; the results were compared to those from innervated hair cells of age-matched larvae in which a sham lesion had been performed (Fig. 2A). Prior to fluorescence-activated cell sorting (FACS), larvae were decapitated to exclude hair cells from the otic vesicle, anterior lateral line and dorsal branch of the pLL. At 1 dpl,

leader axons had only begun to extend along the horizontal myoseptum and most neuromast hair cells were denervated (Fig. S1). By 3 dpl, the majority of neuromast hair cells had been reinnervated and there was functional recovery of swimming behavior. Hair cells were sorted based on size and GFP expression (Fig. S2A), and their identity was confirmed *post hoc* in the bulk sequencing data by cross-referencing the most-expressed genes to a single-cell RNAseq dataset of the neuromast (Fig. S2B,C) (Baek et al., 2022).

Upon dimensionality reduction of the data through principal component analysis, hair cell samples from larvae at 1 dpl clustered separately from age-matched control samples along the first principal component axis, whereas hair cell samples from larvae at 3 dpl were intermixed with their corresponding age-matched controls (Fig. 2B,D; Fig. S3E). Furthermore, there were more genes differentially expressed in hair cells at 1 dpl compared to 3 dpl (Fig. S3A,B). These summary results imply that there are significant changes in the transcriptome of hair cells in response to denervation that revert after hair cells have been reinnervated.

There was no significant upregulation of genes encoding canonical axon guidance cues in the denervated hair cells, but *semaphorin7a* (*sema7a*), *brain-derived neurotrophic factor* (*bdnf*) and *reticulon 4a* (*rtn4a*) were highly expressed across all hair cell samples (Fig. S4A,B). We observed significant enrichment of synapse and extracellular transport gene-ontogeny sets (GO:0045202, GO:0098984, GO:0099003 and GO:0006858) in denervated hair cells at 1 dpl that, with the exception of the synapse gene set (GO:0045202), was lost by 3 dpl (Fig. 2C,E). The gene set for the biosynthesis of heparan sulfate proteoglycans (GO:0015012), which comprises genes for sulfotransferases, glycosyltransferases and other enzymes that modify heparan sulfate side chains, was also transiently enriched in denervated hair cells. There was additionally a transient upregulation of the genes encoding the secreted HSPG core proteins agrin (*agrn*) and collagen XVIII (*col18a1a*), but not of the transmembrane HSPG core proteins syndecan and glypican (Fig. 2F and Fig. S3C,D).

The enrichment of synapse gene-ontogeny gene sets suggests that synaptic genes are modulated in hair cells following nerve lesion compared to innervated controls. The enrichment of genes encoding secreted HSPGs and their modifying enzymes suggests that these cell-signaling molecules may be influencing pLL axonal guidance. Although *agrn* is expressed throughout hair cells and supporting cells of the neuromast, we did not observe pathfinding defects by afferent axons in *agrn*[p168] mutants (Gribble et al., 2018) after nerve lesion (Fig. S5A,B). We accordingly focused on characterizing the role of *col18a1a* during nerve regeneration.

### *col18a1a* is expressed in the neuromast and in Schwann cells at the locations of axonal defasciculation

To further investigate the timing and location of *col18a1a* expression, we performed RNA-FISH on innervated and denervated neuromasts. We observed diffuse expression of *col18a1a* RNA throughout the innervated neuromast, both in hair cells and in supporting cells (Fig. 3A). We also noted localized expression of *col18a1a* in *sox10*[+] Schwann cells positioned outside the neuromast at the locations where an individual axon defasciculated from the nerve. Although *sox10*[+] Schwann cells lined the entirety of the pLL nerve, we counted up to three Schwann cells expressing *col18a1a* at the point of axonal defasciculation (1.02±0.45, mean±s.d.) (Fig. 3F). We did not observe strong expression of *col18a1a* in the rest of the *sox10*[+] Schwann cells that ensheathe the linear, unbranched segments of the nerve (Fig. S6A-E).

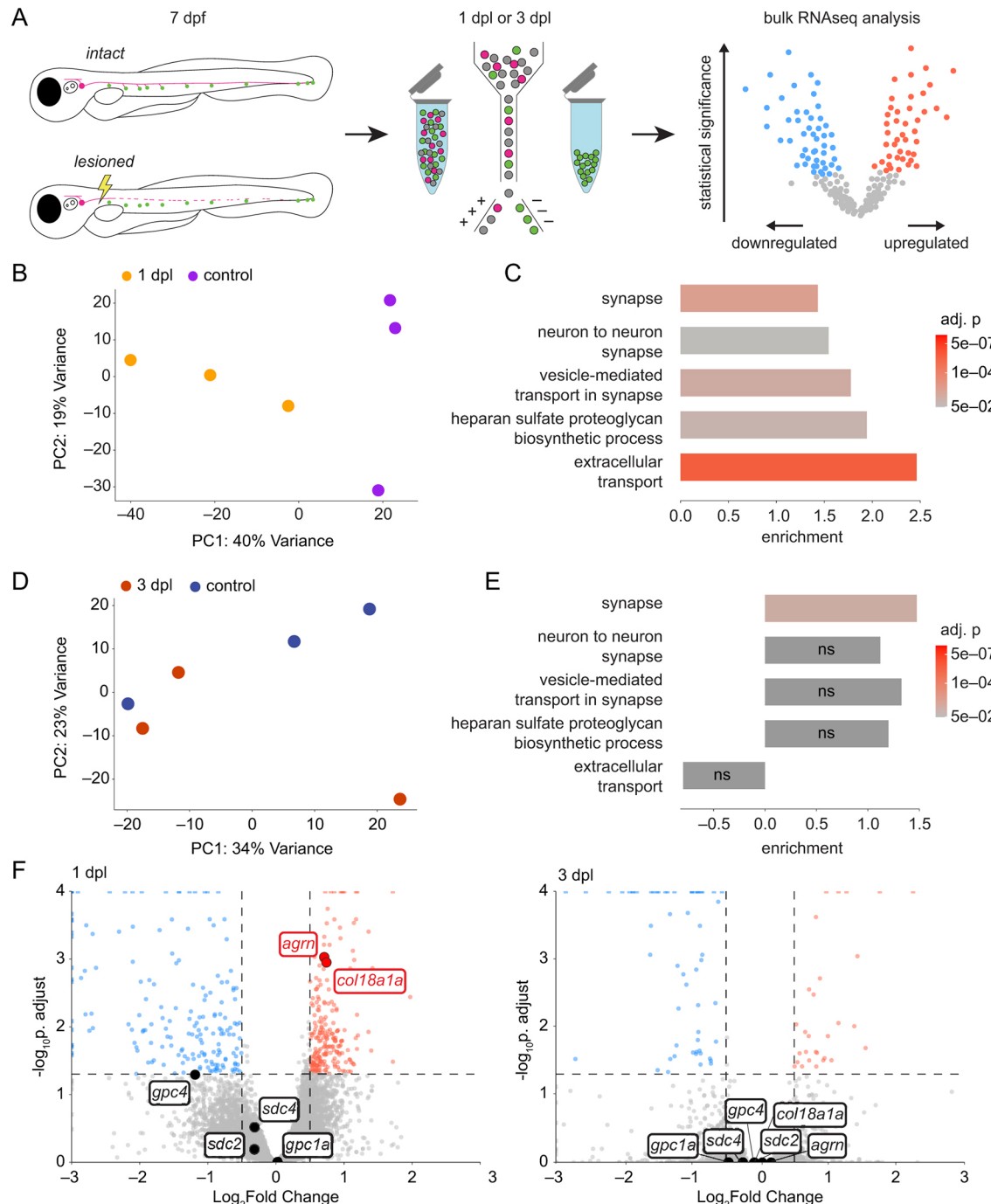

**Fig. 2. Neuromast hair cells transiently change gene expression upon denervation.** (A) A schematic of the hair cell-sequencing workflow. The pLL nerves of *Tg(neurod:tdTomato; myo6b:actb1-EGFP)* larvae were either lesioned or left intact at 7 dpf. Hair cells were isolated at either 1 dpl or 3 dpl and pooled together for bulk sequencing. (B) A principal component analysis plot of samples from denervated hair cells sequenced at 1 dpl (*n*=4058 cells, 4833 cells and 3723 cells) and control hair cells sequenced at 1 day post sham lesion (*n*=5973 cells, 6045 cells and 2744 cells). (C) Gene set enrichment analysis of selected synapse (GO:0045202 and GO:0098984), extracellular transport (GO:0099003 and GO:0006858) and HSPG synthesis (GO:0015012) gene sets in denervated hair cells at 1 dpl compared to age-matched controls. (D) A principal component analysis plot of samples from hair cells sequenced at 3 dpl (*n*=1735 cells, 4363 cells and 3901 cells) and control hair cells sequenced at 3 days post sham lesion (*n*=5123 cells, 2482 cells and 4861 cells). (E) Gene set enrichment analysis of the same GO sets in C, in hair cells at 3 dpl compared to age-matched controls. (F) A volcano plot of differential gene expression in hair cells at 1 dpl (left) or at 3 dpl (right) compared to age-matched controls (red, upregulated; blue, downregulated; gray, not significant), with genes encoding transmembrane and secreted HSPG core proteins highlighted. Statistical significance was set at a *P*=0.05, adjusted for multiple comparisons (horizontal dashed lines). Biological significance is set at 0.5 log₂-fold change (vertical dashed lines). PC, principal component; ns, not significant.

We probed neuromasts at 1 dpl to assess any changes in the spatiotemporal expression of *col18a1a* during pLL nerve regeneration. We were able to document *col18a1a* expression in denervated and recently reinnervated neuromasts within the same larva. We detected the same spatial pattern of *col18a1a* expression in denervated neuromasts and reinnervated neuromasts that we

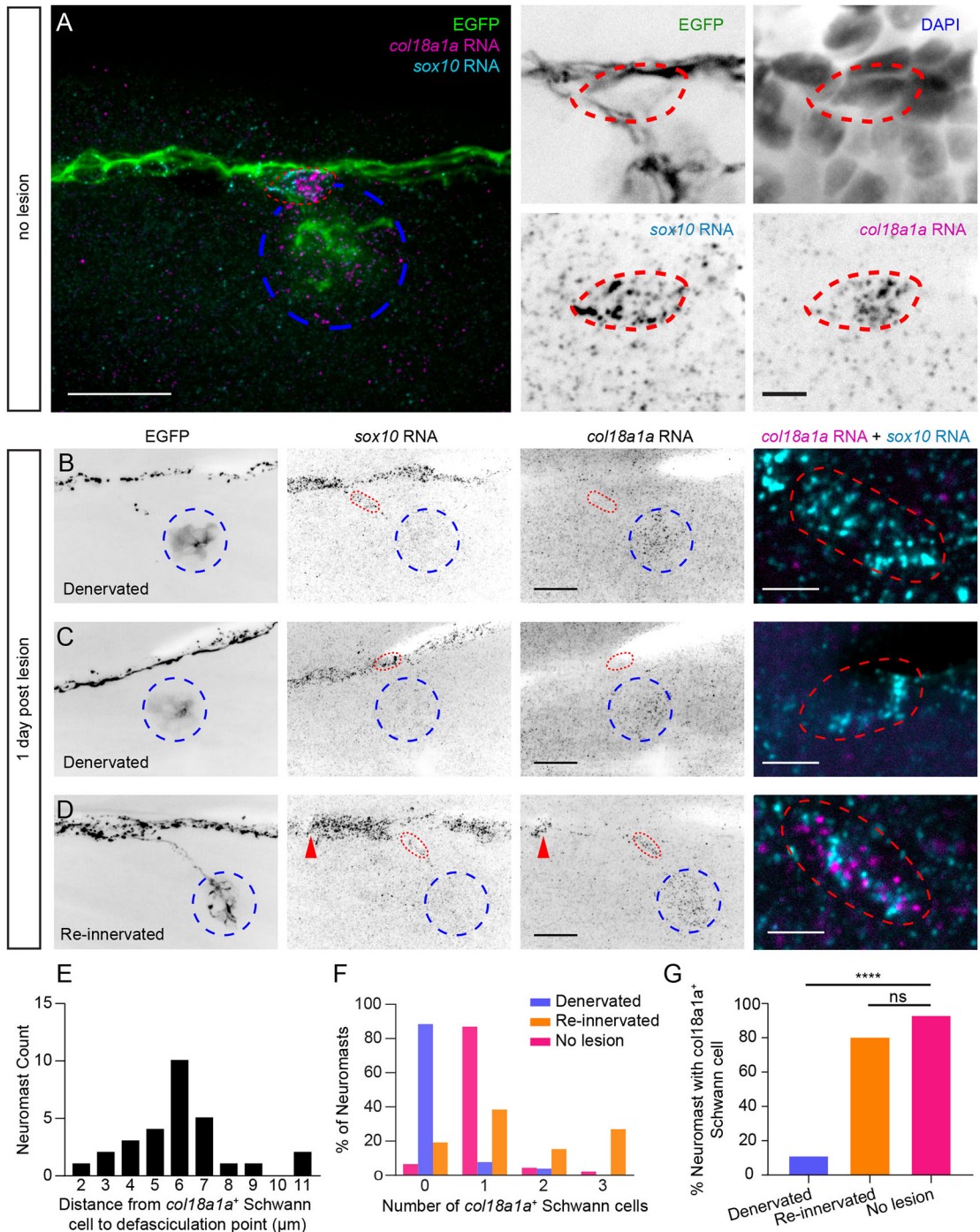

**Fig. 3. Spatiotemporal expression of *col18a1a* RNA in neuromasts and a subset of Schwann cells.** (A) A maximum projection image of a whole neuromast (left) and individual color channels at a single confocal plane at the location of axonal defasciculation (right) of RNA-FISH against *col18a1a* and *sox10* in an 8 dpf *Tg(HGn39d; myo6b:actb1-EGFP)* larva. *col18a1a* is expressed throughout the neuromast (blue dotted outline) and in a single Schwann cell (red dotted outline) adjacent to axonal branching. (B) A denervated neuromast at 1 dpl. The severed nerve is still undergoing Wallerian degeneration. There is expression of *col18a1a* only in the neuromast, and there is a lack of *col18a1a* expression in the Schwann cell located at the original defasciculation point. (C) A denervated neuromast at 1 dpl with a pioneering axon bypassing the neuromast. There is also only *col18a1a* expression in the neuromast and not in the adjacent Schwann cell residing at the original defasciculation point. (D) A re-innervated neuromast. *col18a1a* is expressed in the neuromast and in two Schwann cells adjacent to axonal branching (red arrowheads and red dotted outline). An overlay of *col18a1a* and *sox10* probe signal in the Schwann cells outlined in red is shown on the rightmost panel of B-D. (E) The distance from the point of axonal defasciculation to the nearest *col18a1a*+ Schwann cell. *n*=29 primary, non-terminal neuromasts from 21 larvae. (F) The distribution of the number of adjacent *col18a1a*+ Schwann cells in innervated control neuromasts (no lesion), denervated neuromasts and re-innervated neuromasts. (G) The percentage of denervated, re-innervated and control neuromasts with at least one adjacent *col18a1a*+ Schwann cell. *n*=26 denervated neuromasts from 19 larvae, 26 re-innervated neuromasts from 19 larvae and 46 control neuromasts from 27 larvae. \*\*\*\**P*<0.0001 by Fisher's exact test. ns, not significant. Scale bars: in A, 20 μm (left) and 5 μm (right); in B-D, 20 μm (single channels) and 5 μm (overlay).

observed in control neuromasts. The high variability of probe intensity and background signal prevented us from detecting significant differences in fluorescence intensity between neuromasts at different timepoints following nerve lesion. However, we did note a decrease in the number of *col18a1a*⁺ Schwann cells adjacent to denervated neuromasts (0.15±0.46, mean±s.d.) (Fig. 3B,C,F). In neuromasts that had been reinnervated, we again detected the presence of at least one *col18a1a*⁺ Schwann cell at the axon branch point (1.5±1.11, mean±s.d.) (Fig. 3D,F,G). Based on these results, we speculated that the dynamic expression of *col18a1a* by specialized Schwann cells, located at specific choice points along the nerve, may be influencing axonal pathfinding during nerve regeneration.

To further investigate the relationship between pLL axons and the ECM at branching points, we used a *Tg(krt19:col1α2-GFP; hsp70: NTR2.0-2A-mCherry en.Sill1)* transgenic line, henceforth referred to as *Tg(krt19:col1α2-GFP; sill1:mCherry)*. This line expresses the fluorescent protein mCherry in lateral line afferents (Pujol-Marti et al., 2012) and a secreted collagen I-GFP fusion protein that incorporates into a collagen I mesh directly underneath the epidermal basement membrane (Morris et al., 2018). At 5 dpf, we observed well-circumscribed gaps in the collagen I mesh through which defasciculated axons passed to innervate hair cells of primary neuromasts residing in the epidermis (Fig. 4A; Movie 2). In secondary neuromasts, which are deposited by a second migrating primordium and lie farther ventral from the nerve compared to primary neuromasts, defasciculated axons travelled within the collagen I mesh, rather than through a circumscribed gap (Fig. S7A). Although the entrance to the collagen I mesh lay where axons defasciculated from the nerve and the exit corresponded to the location of the neuromast in the epidermis, these locations were variable and difficult to locate precisely. For this reason, we restricted further analysis to primary neuromasts, for which a circumscribed gap could always be clearly defined.

In contrast to the axon bundle, which coursed medial to the epidermis (Fig. 4A), neuromast cells and adjacent interneuromast

cells labeled in *Tg(she:H2A-mCherry; krt19:col1α2-GFP)* fish were positioned lateral to the basement membrane (Fig. 4B). To localize *col18a1a* expression in relation to the ECM, we performed RNA-FISH on fixed 8 dpf *Tg(krt19:col1α2-GFP)* larvae. Although the fixation and *in situ* protocol distorted the ECM, we could nonetheless observe a single *col18a1a*⁺ cell directly medial to the collagen I mesh and adjacent to a gap. In comparison, the *col18a1a* expression in the neuromast occurs lateral to the basement membrane (Fig. 4C). The constrained expression of *col18a1a* across the epidermal boundary layer, in proximity to the gaps through which defasciculated axons pass, further indicates a role in axonal guidance.

## Branching axons of the regenerating pLL nerve extend through pre-formed gaps in the extracellular matrix

The gaps that permit extension of defasciculated axons into the epidermis are a consequence of basement membrane remodeling that occurs during development (Raphael et al., 2010). At 5 dpf, the gaps had an area of 12.47±5.46 μm² (mean±s.d.) and were positioned close to the nerve, within 4.24±1.95 μm (mean±s.d.) of axon defasciculation (Fig. 5B,C). We further investigated the role of these gaps during nerve regeneration. We imaged the innervation of primary neuromasts through these gaps in 5 dpf *Tg(krt19:col1α2-GFP; sill1:mCherry)* larvae, lesioned the nerve and reimaged the same gaps at 1 dpl and 3 dpl (Fig. 5A′-A″). In 31 out of 32 larvae, we observed regeneration through the same ECM gap through which the axons extended before lesioning. In the one exceptional case, axons traversed a different pre-existing gap in the ECM. Gaps remained the same size and shape during the 3-day course of nerve regeneration (Fig. 5C,D), and we did not observe the formation of any new gaps in this interval.

To determine whether axons would extend into the epidermis through any gap in the ECM, we punctured the epidermal boundary layer with a glass microneedle somewhere along the path of the regenerating nerve. Each puncture created an ~30 μm-wide gap in

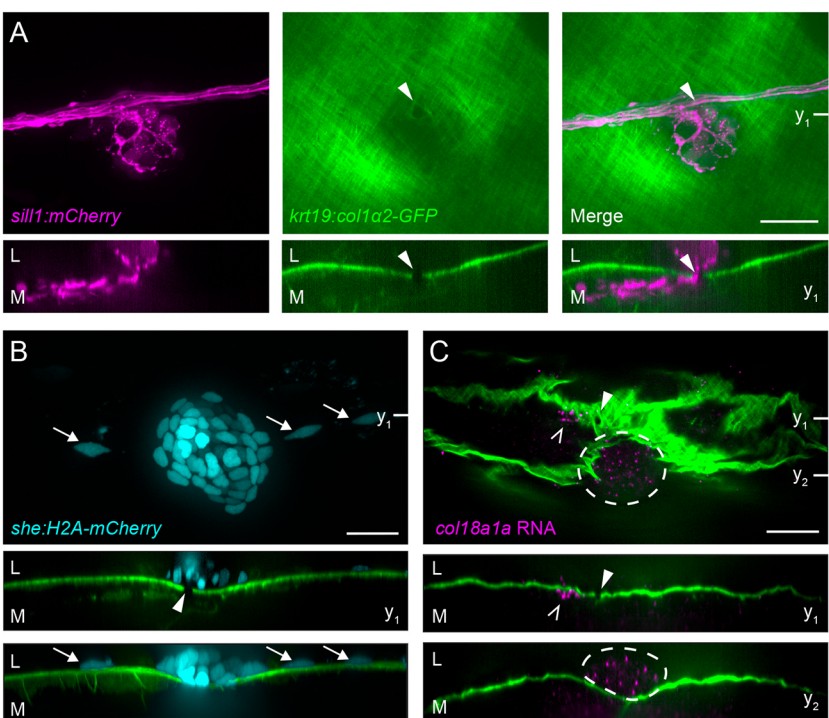

**Fig. 4. Axons cross gaps in the epidermal boundary layer to innervate neuromast.** (A) A parasagittal projection (top) and a coronal slice (bottom) taken at the indicated plane (y₁) of a confocal stack imaged in a live 5 dpf *Tg(krt19:col1α2-GFP; sill1:mCherry)* larva. Individual afferents labeled by the *sill1* enhancer (magenta) defasciculate from the nerve and pass through a gap (white arrowhead) in the collagen I matrix (green). Arrowheads indicate ECM gap. (B) A sagittal projection (top), coronal slice (middle) taken at the indicated plane (y₁) and a coronal projection (bottom) of a confocal stack imaged in a live 4 dpf *Tg(krt19:col1α2-GFP; she:H2A-mCherry)* larva. Neuromast (cyan) and adjacent interneuromast cells (cyan, white arrows) reside in the epidermis, lateral to the collagen I matrix and gap. mCherry fluorescence intensity gamma corrected. Arrowhead indicates ECM gap. (C) An individual sagittal slice (top) and coronal slices (middle and bottom) taken at the planes indicated (y₁ and y₂) of RNA-FISH against *col18a1a* in a fixed, 8 dpf *Tg(krt19:col1α2-GFP)* larva. A single *col18a1a*⁺ cell resides medial to the epidermal boundary layer (white carets) and adjacent to the ECM gap (white arrowheads). Cells of the neuromast (white dotted outline) are also *col18a1a*⁺ and reside lateral to the epidermal boundary. L, lateral; M, medial. Scale bars: 20 μm.

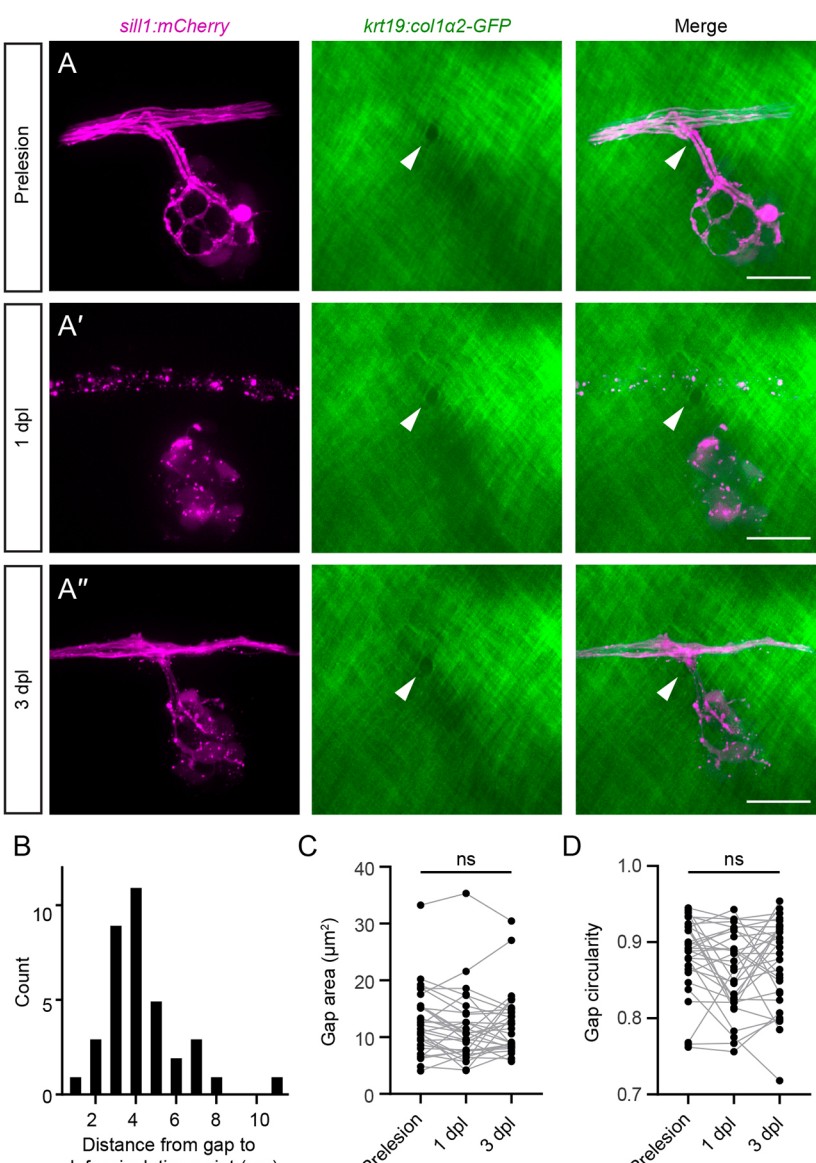

**Fig. 5. Defasciculation through a boundary gap during axon regeneration.** (A) Before nerve lesion, a primary neuromast in a 5 dpf *Tg(krt19:col1α2-GFP; sill1:mCherry)* larva was innervated through a gap in the ECM (white arrowheads). (A′) While axons underwent Wallerian degeneration at 1 dpl, the gap in the ECM remained unchanged. (A″) By 3 dpl, axons had re-innervated the neuromast by defasciculating through the original ECM gap. The location, morphology and size of the gap did not change after nerve lesion. (B) The distribution of distances between the ECM gap and the point of axon defasciculation in 5 dpf larvae. (C,D) The gap areas (C) and gap circularity (D) before and after nerve lesion. *n*=36 zebrafish. ns, not significant. Scale bars: 20 µm.

the ECM. There was full extension of an axon into the epidermis through this artificial gap in only one of 33 larvae imaged. For the other 32 larvae, the majority of axons remained tightly fasciculated beneath the epidermal boundary, with the occasional axon stalled along the edges of the gap (Fig. 6A).

In some larvae, the puncture was made along the path of the regenerating nerve, within 50 µm of a neuromast. In these cases, we could test whether the regenerating axons would pass through the original gap or the artificial gap to reinnervate the neuromast. Because of their size and irregular borders, the artificial gaps were easy to distinguish from the smaller more-circular gaps that naturally formed during development. We found that axons preferred to reinnervate neuromasts through the natural gaps compared to the artificial gaps (10 out of 11 larvae). This was true whether the artificial gap was located posterior (five of five larvae; Fig. 6B; Movie 3) or anterior (five of six larvae; Fig. 6C) to the natural gap. The preference of an axon to defasciculate through the natural gap in the ECM suggests that there are attractive properties unique to this location that draw growth cones into the epidermis.

## *col18a1a* mutants have aberrant axon pathfinding during pLL nerve regeneration

Based on the spatiotemporal expression pattern of *col18a1a*, we hypothesized that this extracellular HSPG influences the defasciculation of individual axons. We generated a stable *col18a1a* mutant line by CRISPR/Cas9 gene editing. We selected mutants with a 2 bp deletion and a frameshift resulting in a premature stop codon in an exon common to all three isoforms of *col18a1a* (Fig. S8A,B). These mutants were bred into a *Tg(HGn39d; myo6b:actb1-EGFP)* background so as to have an afferent axon and a hair cell marker. There was a significant decrease in *col18a1a* RNA labeling in the neuromast hair cells of homozygous mutants compared to wild-type siblings, as well as a loss of signal at normal defasciculation points, suggesting nonsense-mediated decay of mutant transcripts (Fig. 7A,B). Although innervation of neuromasts appeared grossly normal in homozygous mutants, mutant nerves displayed higher tortuosity (Fig. 7C,D) and were less tightly packed together compared to those of siblings (Fig. 7E).

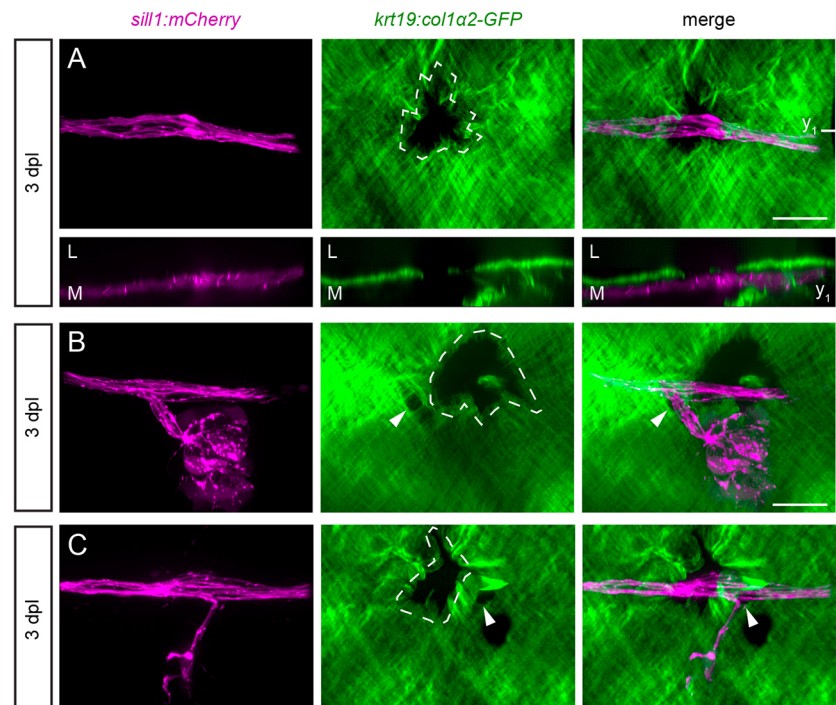

**Fig. 6. Axons do not defasciculate through artificial gaps in the ECM.** (A) A parasagittal projection (top) and coronal slice at the plane indicated ($y_1$) (bottom) of a regenerated nerve bypassing a puncture made in the ECM (dotted white outline) in an 8 dpf *Tg(krt19:col1α2-GFP; sill1:mCherry)* larva. (B) In another larva, the re-innervation of a neuromast proceeded through a natural ECM gap (white arrowheads) rather than through a puncture made immediately posterior to the gap. (C) Re-innervation of a neuromast in a different larva similarly bypassed a puncture made immediately anterior to the natural ECM gap (white arrowheads). L, lateral; M, medial. Scale bars: 20 μm.

We checked for abnormalities in *col18a1a^{ru703/ru703}* mutants following pLL nerve lesion and axon regeneration. Although there was reinnervation of all neuromasts, we observed individual axons defasciculating from the regenerated axon bundle at inappropriate locations. These axons would branch from the nerve where there was no nearby neuromast (Fig. 7F). These additional branches were able to cross the basement membrane through small openings that were distinct from the native gaps adjacent to neuromasts. These branches extended away from the nerve, lateral to the epidermal boundary layer (Fig. S8C). While we found examples of inappropriate defasciculation at different locations along the trunk, we noted that many abnormal branches were in somites adjacent to secondary neuromasts (Fig. 7G). We also observed mutant larvae in which a lone axon extended past the terminal neuromast, into the developing tailfin (Fig. 7H). We observed significantly more mutant larvae with aberrant axon pathfinding compared to control siblings (Fig. 7I). Our results indicate that, although *col18a1a* is not required for pLL nerve regeneration or hair cell reinnervation, it plays an auxiliary role in refining axon defasciculation and subsequent pathfinding to appropriate locations.

## DISCUSSION

Although the path that severed axons follow resembles the path established during development, there are key differences in how the growth cones reach their targets. The innervation pattern of neuromasts deposited during development reflects the chronology of neuron differentiation in the ganglion. Peripheral axons of early-differentiating neurons contact the primordium lateral to the epidermal basement membrane and project to the caudal end of the fish, whereas axons of later-differentiating neurons trail behind the primordium and innervate more rostral neuromasts (Pujol-Martí et al., 2010; Sato and Takeda, 2013). In contrast, during regeneration, axons must advance medial to the epidermal basement membrane in the absence of a primordium and instead extend along bands of Büngner (Xiao et al., 2015). Leader axons

bypass denervated neuromasts to reach the caudal end of the fish (Fig. 1B). Follower axons have a simpler task of pathfinding by growth along the shaft of a leader. There is great plasticity in the rewiring of neuromasts following nerve lesion, with individual axons able to reinnervate different neuromasts from those that they originally innervated (Ceci et al., 2014; Lozano-Ortega et al., 2018). This pattern implies that the decision to defasciculate from the axon bundle is not hard-wired into an axon and is biased by external signaling cues.

To identify these cues, we first sequenced denervated hair cells in bulk to determine if there were transcriptional changes associated with axon guidance. We observed that hair cells transiently changed their gene expression profile in response to denervation. Although we did not see an upregulation of canonical axon guidance cues, we observed an upregulation of *col18a1a*, which encodes a secreted HSPG. HSPGs have an extensive role in neuronal migration, axon guidance and synaptogenesis during nervous system development (Poulain and Yost, 2015). Once secreted from the cell, HSPGs alter the structure of basement membranes and provide storage depots for growth factors and other diffusible cues (Sarrazin et al., 2011). Although we saw additional branching at inappropriate locations during pLL nerve regeneration in our *col18a1a* mutants, we nevertheless observed proper axon defasciculation toward neuromasts. This behavior suggests an accessory role for collagen XVIII in patterning and refining axon branching rather than a primary guiding role. HSPGs can bind canonical axon guidance cues, such as Netrins, Slits, and Ephrins, and modulate their activity in the extracellular space (Hussain et al., 2006; Irie et al., 2008; Serafini et al., 1994).

We were surprised to find 1-3 *col18a1a^+* Schwann cells lying outside a neuromast, at the location of axon defasciculation. The expression of *col18a1a* in these Schwann cells was dynamic over the course of nerve regeneration, disappearing directly after nerve lesion and re-appearing to coincide with reinnervation. The initial downregulation of *col18a1a* expression would correspond with the immediate de-differentiation of Schwann cells that occurs during

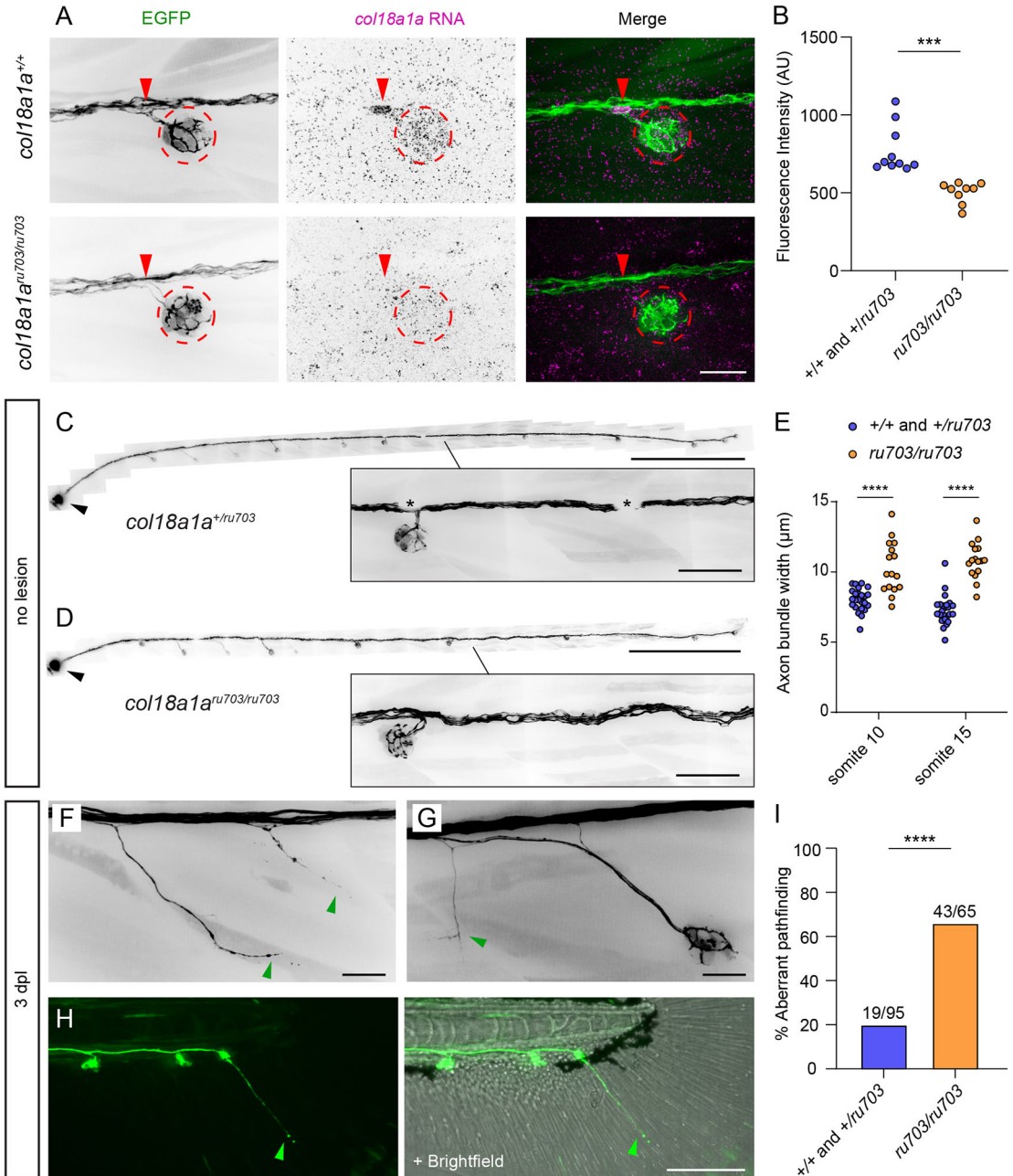

**Fig. 7. *col18a1a* mutants have aberrant axon pathfinding during pLL nerve regeneration.** (A) RNA-FISH against *col18a1a* in 5 dpf *Tg(HGn39d; myo6b: actb1-EGFP)* wild type and *col18a1a^ru703/ru703^* mutants. In mutants, there was decreased expression in the neuromast (dotted red outline) and at the axonal defasciculation point (red arrowheads). (B) The average fluorescence intensity of the *col18a1a* RNA probe signal in the hair cells of the neuromasts in *col18a1a^ru703/ru703^* mutants compared to siblings. *n*=9 mutants, *n*=10 siblings. (C) The peripheral segment of the pLL nerve in an 8 dpf *col18a1a^+/ru703^* *Tg(HGn39d; myo6b:actb1-EGFP)* larva. Asterisks indicate melanocytes overlying the nerve. (D) In the peripheral segment of the pLL nerve in an 8 dpf *col18a1a^ru703/ru703^* *Tg(HGn39d; myo6b:actb1-EGFP)* larva, the nerve is less tightly packed and follows a more tortuous path (inset). The pLL ganglion is marked by a black arrowhead in C and D. (E) The width of the axon bundle in 5 dpf *col18a1a^ru703/ru703^* mutants compared to control siblings at the level of the 10th and 15th somites. *n*=16 mutants, *n*=24 siblings. (F) Aberrant defasciculation of axons (green arrowheads) in a mutant larva after nerve regeneration. (G) Normal re-innervation of a secondary neuromast accompanied with inappropriate branching (green arrowhead). (H) Extension of a lone axon (green arrowhead) past the terminal neuromast in the developing tailfin. (I) The percentage of *col18a1a^ru703/ru703^* mutant larvae with aberrant axon pathfinding at 3 dpl compared to control siblings. *n*=95 wild-type/heterozygous larvae, 65 mutant larvae. (B) \*\*\**P*<0.001, unpaired *t*-test. (E) \*\*\*\**P*<0.0001, unpaired *t*-test. (I) \*\*\*\**P*<0.0001, chi-squared test. Scale bars: 20 μm in A,F,G; 500 μm in C,D (50 μm in insets); 100 μm in H.

Wallerian degeneration and the clearance of axonal debris (Ceci et al., 2014; Xiao et al., 2015). The re-emergence of *col18a1a* expression in a small subset of Schwann cells occurs between the passage of a pioneering axon and the reinnervation of the neuromast by a defasciculated follower axon, as Schwann cells re-differentiate following the onset of axon regeneration.

We speculate that collagen XVIII, secreted by the neuromast and by specialized Schwann cells, may scaffold with axon guidance cues to create a signaling pathway from one side of the epidermal basement membrane to the other (Fig. 8). Potential candidates for complexed guidance cues include: GDNF, a neurotrophic factor necessary for the association of towed axons with the primordium

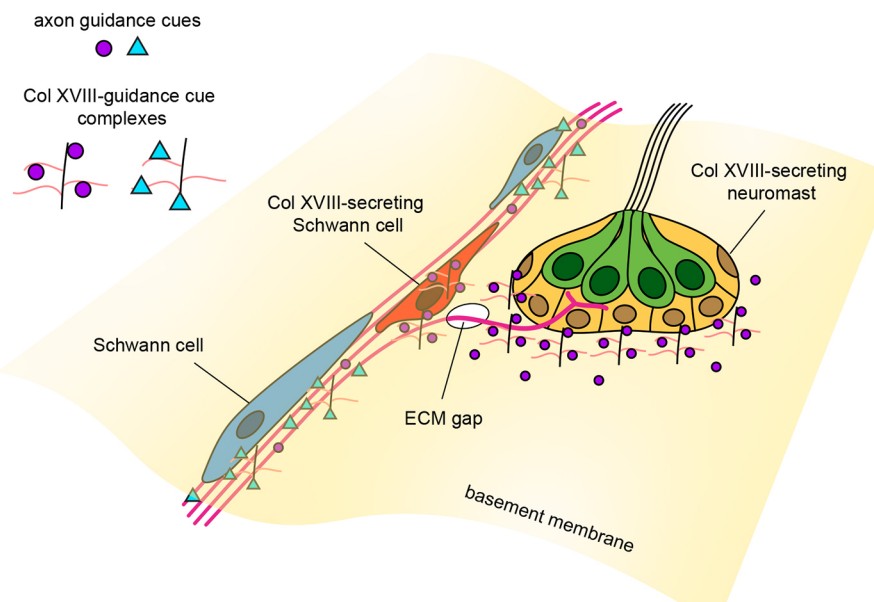

**Fig. 8. A scaffolding model for patterning guidance cues in the basement membrane.** Collagen XVIII may indirectly affect the precision of axon guidance by scaffolding with different axon guidance cues in the basement membrane. Depending on the identity of the guidance cues and where they are localized in the extracellular space, the proposed signaling complex could have pro-bundling and pro-branching effects at different locations along the nerve. Axon guidance cues could be complexed directly to the collagen XVIII core protein or to heparan sulfate side chains.

(Schuster et al., 2010); semaphorin 7a, a hair cell-derived chemoattractant (Dasgupta et al., 2024); BDNF, a neurotrophin expressed by hair cells and crucial for maintaining afferent innervation (Aragona et al., 2022; Mo and Nicolson, 2011); and reticulon 4/nogo-A, which has permissive axon guidance activity in zebrafish, unlike its inhibitory effect in mammals (Abdesselem et al., 2009; Welte et al., 2015).

In addition to excessive aberrant branching in *col18a1a* mutants, we also observed nerve-wide phenotypic differences in which the nerve follows a more tortuous path and is less tightly bundled together, suggesting that collagen XVIII may have a pro-fasciculation effect along the nerve. The proposed scaffolding model is one way of reconciling the multiple roles collagen XVIII may have. By complexing with different cues, collagen XVIII could have varied effects on axons at different positions along the nerve. While the *col18a1a* gene is necessary for maintaining the spatial fidelity of axon navigation, the exact context-dependent effects of the translated protein remain an area of further study.

Crossing the basement membrane represents a unique challenge to regenerating pLL axons. During development, the ECM is actively remodeled by Schwann cells to position the axon bundle beneath the epidermis and thereby isolates the nerve from post-embryonic ventral migration of neuromasts (Raphael et al., 2010). This remodeling leaves small gaps in the ECM near the nerve through which defasciculated neurites enter the epidermis. Although these gaps can be seen in serial electron micrographs of the neuromast (Dow et al., 2018; Odstrcil et al., 2022; Raphael et al., 2010), there have been no studies visualizing these gaps in a live zebrafish. Using a transgenic line that expresses a collagen I-GFP fusion protein that incorporates into the epidermal boundary, we were able to visualize ECM gaps over the course of nerve regeneration. In contrast to the active remodeling of the ECM that occurs during development, axons re-extended through pre-formed gaps that remained static over the 3-day course of regeneration. Although axon guidance has traditionally been viewed as a product of chemical signaling, it is increasingly apparent that mechanical signaling in the form of stiffness gradients also regulates pathfinding (Koser et al., 2016). One possibility is that a gap in the ECM is sensed by the growth cone as a change in substrate

stiffness and itself induces defasciculation. We have shown that the presence of an artificially made gap in the ECM is not enough to induce defasciculation of axons into the epidermis, and that axons prefer to grow through the smaller original gap rather than through the larger artificial one. This behavior implies that the biochemical and physical properties of the native gap are necessary for attracting cut axons into the epidermis.

Once inside the neuromast, growth cones must still distinguish hair cells from supporting cells, likely through contact with membrane-anchored synaptic markers on the basal surfaces of hair cells. Our bulk-sequencing data from denervated hair cells provide a good starting point for identifying candidate markers. Upon denervation, we observed upregulation of two genes that encode hair cell-specific synaptic adhesion molecules, *nrxn3b* and *lingo3a* (Fig. S9A-C). Neurexin3b has been shown to be essential for pairing pre- and post-synapses in neuromast hair cells (Jukic et al., 2024). Although Lingo3a is an uncharacterized orphan receptor, the Lingo protein family has been proposed to regulate nervous system development and axonal survival, as do other leucine-rich repeat transmembrane proteins (Guillemain et al., 2020).

In addition to restoring hair cell ribbon synapses with afferent fibers, there must also be recovery of modulatory efferent innervation. There are excitatory dopaminergic inputs that indirectly signal to hair cells, and inhibitory cholinergic inputs that directly synapse with hair cells (Manuel et al., 2021; Odstrcil et al., 2022; Toro et al., 2015). Both dopaminergic and cholinergic efferents do not associate with the primordium, and instead follow the established path of towed afferents (Manuel et al., 2021; Toro et al., 2015). We assume that the regrowth of efferent fibers also likely depends on their association with regenerated afferent fibers, which greatly outnumber efferents in the nerve and in the neuromast (Dow et al., 2018). Once in the neuromast, different synaptic adhesion molecules on the hair cell must sort afferents to ribbon synapses and inhibitory efferents to post-synaptic membrane cisterns with acetylcholine receptors. Although afferent innervation is selective for hair cells of an identical orientation within a neuromast (Dow et al., 2018; Faucherre et al., 2009; Lozano-Ortega et al., 2018; Nagiel et al., 2008), efferent innervation

is not, suggesting further differences in how the two synapses are re-established. Agrin, another secreted HSPG that was upregulated in our bulk sequencing data, has a conserved role in clustering acetylcholine receptors and facilitating synaptogenesis at neuromuscular junctions through the Agrn/Lrp4/MuSK signaling pathway (Gribble et al., 2018; Kim et al., 2008; Zhang et al., 2008). While we did not see any gross defects in the afferent re-innervation of hair cells in *agrn^p168* mutants in which the neural z-isoform is affected (Gribble et al., 2018), we cannot rule out impaired reinnervation by cholinergic efferents, or a role for a different *agrn* isoform. Further work must be carried out to elucidate the function of Agrin in neuromast reinnervation, and, more generally, the differences between efferent and afferent regeneration.

The simplicity and accessibility of the zebrafish posterior lateral line makes it an ideal model for studying the basic mechanisms of peripheral nerve regeneration and target selection. We imaged distinct gaps in the ECM through which axons selectively regenerate, and we identify *col18a1a* as an important gene for maintaining the precision of axonal pathfinding. A detailed understanding of axon guidance *in vivo* and the reformation of the sensory synapse has the potential to reveal therapies driving neural regeneration in otherwise permanently damaged sensory organs.

## MATERIALS AND METHODS
### Zebrafish strains
Zebrafish work was performed in accordance with animal protocol 22081-H reviewed and approved by The Rockefeller University's Institutional Animal Care and Use Committee. Adult zebrafish (*Danio rerio*) were maintained under standard conditions (Nusslein-Volhard and Dahm, 2002). Naturally spawned and fertilized eggs were maintained in system water treated with 0.5 mg/l Methylene Blue (blue water) at 28.5°C on a 14-/10-h light/dark cycle. Experiments were performed on 4-10 dpf larvae, before sex was determined.

The following previously described zebrafish transgenic and mutant lines were used: *Tg(cntnap2a)^nkhgn39dEt* (Nagayoshi et al., 2008), *Tg(myo6b: actb1-EGFP)^vo8Tg* (Kindt et al., 2012), *Tg(neurod:tdtomato)^vo12Tg* (Toro et al., 2015), *Tg(krt19:colIα2-GFP)^zf2175Tg* (Morris et al., 2018), *Tg(she: H2A-mCherry)^psi57Tg* (Peloggia et al., 2021) and *agrn^p168* (Gribble et al., 2018). The *Tg(hsp70:NTR2.0-2A-mCherry en.Sill1)* line was created using the Gateway-based Tol2 kit (Kwan et al., 2007). Expression plasmids were created by combining *p5E-hsp70* (Kwan et al., 2007), *pME-NTR2.0-2A-mCherry* (Sharrock et al., 2022), *p3E-Sill1* (Pujol-Marti et al., 2012) and *pDestTol2pA2* (Kwan et al., 2007) plasmids. Verified plasmid DNA was injected at a concentration of 30 ng/µl into single-cell AB embryos along with 125 ng/µl of Tol2 transposase mRNA. The mutant *col18a1a^ru703* line was created through CRISPR/Cas9 gene editing of wild-type embryos. Adults with a 2 bp deletion in exon 2 (201, 203 transcript) or exon 3 (202 transcript) of the *col18a1a* gene (ENSDARE00000715935) were pooled and outcrossed to *Tg(HGn39d; myo6b:actb1-EGFP)* fish to generate heterozygotes identified by tail-clip genotyping with primers: *col18a1a^ru703* forward, 5′-GCGTCTCCAAAGTCTTCGAC-3′; and *col18a1a^ru703* reverse, 5′-GCGCTCGAAATTGACAACCT-3′. Similarly, *agrn^p168* mutants were outcrossed to *Tg(HGn39d; myo6b:actb1-EGFP)* fish to generate heterozygotes identified by tail-clip genotyping with primers: *agrn^p168* forward, 5′-CAATGGTCAGAAGACAGACGG-3′; and *agrn^p168* reverse, 5′-GGCTCCACTGTATATTATGCTGC-3′.

### Lateral line nerve lesion and live imaging
Double transgenic *Tg(myo6b:actb1-EGFP; HGn39d)* larvae were anesthetized at 4 dpf in 600 µM tricaine (MS-222 tricaine methanesulfonate, Syndel) and embedded in 1% low melting point agarose for laser lesioning of the afferents of the right pLL nerve. All imaging was performed on a microlens-based, super-resolution confocal imaging system (VT-iSIM, Biovision Technologies) fixed to an Olympus IX81 frame and using a UPlanSApo 60×1.3 NA silicone oil objective

(Olympus). By convention, all displayed parasagittal images are oriented with the anterior of the fish to the left, the posterior of the fish to the right, the dorsal surface of the fish to the top and the ventral surface of the fish to the bottom. All displayed coronal images are oriented with the anterior of the fish to the left, the posterior of the fish to the right, the lateral surface of the fish to the top and the midline to the bottom. A 20 µm segment of the axon bundle at the 4th somite was irradiated at three different confocal planes with an attached iLas Pulse laser system (Gataca Systems, France) equipped with a 355 nm laser, and complete transection of the bundle was confirmed 1 h after ablation. The larvae were immersed in an imaging solution of 120 µM tricaine, 0.4 mg/ml pancuronium bromide (P1918, Sigma-Aldrich), and 1 mM sodium-L-ascorbate (11140, Sigma-Aldrich) diluted in system water for extended timelapse imaging of over 48 h. Confocal stacks of the L1 neuromast were acquired at 1 µm intervals along the sagittal axis every 40 min.

The ErbB inhibitor AG1478 (658548, Sigma-Aldrich) was used to pharmacologically block Schwann cell signaling during axon bundle regeneration. The right pLL nerve was lesioned as described in 4 dpf *Tg(HGn39d)* larvae and the larvae were freed from the agarose and left to recover in either 1-4 µM AG1478 in 1% DMSO (Invitrogen; D12345) or only 1% DMSO in system water. Larvae were remounted for imaging 2 dpl, and the somite reached by the leading axon was recorded.

### Bulk RNA sequencing of denervated hair cells
The right and left pLL nerve was lesioned in 100-150 7 dpf *Tg(neurod: tdtomato; myo6b:actb1-GFP)* larvae. For sham lesions, tank siblings were exposed on both right and left sides to a UV LED for 30 s. Larvae were left to recover in blue water for either 1 or 3 days after lesion before tissue dissociation. We followed a modified neuromast tissue dissociation protocol (Baek et al., 2022) to generate a crude cell suspension for fluorescence-activated cell sorting (FACS). Larvae were anesthetized in 600 µM tricaine and decapitated with a fresh razor blade immediately anterior to the swim bladder to exclude hair cells from the anterior lateral line, otic vesicle and dorsal branch of the pLL. The remaining larval trunks were rinsed with ice-cold 1×DPBS (Gibco, 14190144)/0.04% BSA (03117332001, Roche) and transferred to a 15 ml polypropylene conical tube in 4.5 ml of ice-cold 0.25% trypsin-EDTA (Gibco; 25200056). The trunks were dissociated by trituration with a 1 ml pipette tip for 4 min on ice. The subsequent crude cell suspension was filtered through a 70 µm pore cell strainer into a 15 ml polypropylene round-bottom centrifuge tube. We centrifuged the cells at 800 $\boldsymbol{g}$ for 5 min at 4°C. The supernatant was carefully removed from the cell pellet and the pellet was resuspended in ice-cold 1×DPBS/0.04% BSA before re-centrifugation at 800 $\boldsymbol{g}$ for 5 min at 4°C. This step was repeated once more, and the cell pellet resuspended in 500 µl of 1×DPBS/0.04% BSA. This suspension was filtered through another 70 µm pore filter into a 1.5 ml screwcap tube with 100 units of Protector RNase Inhibitor (3335399001, Roche). The total time between larvae decapitation and cell sorting was kept to under 90 min to minimize hair cell death.

GFP-positive cells were sorted at the Rockefeller University Flow Cytometry Resource Center with a BD FACSAria II (BD Biosciences) using a 100 µm nozzle at 20 psi. The 405, 488 and 561 nm laser excitation lines with 450/40 nm, 515/20 nm and 586/15 nm emission filters, respectively, were used for collection. For every sorting session, a fresh control sample of GFP-negative cells was used to establish gating parameters. GFP-positive hair cells were sorted based on size and GFP fluorescence directly into RLT+ cell lysis buffer (QIAGEN) with 1% β-mercaptoethanol. Total sorting time was limited to 30 min to limit hair cell death. Total RNA was extracted after sorting using the RNeasy Plus Micro Kit (QIAGEN) and stored at −80°C until sequencing. Hair cell RNA from three technical replicates was collected for each condition (1 day post nerve lesion, 1 day post sham lesion, 3 days post nerve lesion and 3 days post sham lesion).

The yield and quality of isolated RNA was measured with a bioanalyzer (Agilent 2100). All samples had an RNA integrity score (RIN) greater than 8.0 and were subsequently sequenced by the Rockefeller University Genomics Resource Center. 1 ng of total RNA was used to generate full-length cDNA using the Clontech Smart-Seq v4 Ultra Low Input RNA kit (634888, Clontech). 1 ng of cDNA was then used to prepare libraries using the Illumina Nextera XT DNA sample preparation kit (FC-131-1024,

Illumina). Libraries generated from hair cells 1 day post nerve lesion and control hair cell samples 1 day post sham lesion were sequenced together on an Illumina NextSeq500 with single end 75 bp reads. Libraries generated from hair cells 3 days post nerve lesion and control hair cell samples 3 days post sham lesion were sequenced together on an Illumina NextSeq 2000 with single end 100 bp reads. All samples were sequenced to a depth of ∼66 million reads.

Quality control and processing of the raw bulk RNA sequencing data was performed by the Rockefeller Bioinformatics Resource Center using a standardized bulk sequencing analysis pipeline programmed in R. Briefly, reads were aligned to the zebrafish reference genome (GRCz11) using the Rsubread package, and transcript counts quantified with Salmon and the tximport package. Quality control metrics were assessed with the picard package. Batch correction for combined PCA analysis was performed using the limma package. Processed data were normalized and statistical testing was performed using the DESeq2 package. Only genes with normalized transcript counts above 500 were included in volcano plots and MA plots. Gene set enrichment analysis was performed using the fgsea package (Korotkevich et al., 2021 preprint). All raw sequencing data have been deposited in GEO) under accession number GSE297706. Detailed step-by-step workflows, quality control metrics, and additional R code used for the processing of bulk sequencing data in this study are publicly available (https://github.com/rsroy27/CollagenXVIII_2025_Manuscript). Gene lists and normalized transcript counts for each sample are provided in Tables S1 and S2.

### RNA fluorescence *in situ* hybridization

RNA fluorescence *in situ* hybridization was performed on whole larvae using standardized hybridized chain reaction (HCR) v3 and v4 protocols, reagents, and probes (Molecular Instruments). Larvae were fixed in 4% paraformaldehyde for 24 h at 4°C either 1 day post nerve lesion or with the nerve left intact. Fixed larvae were washed in PBS and dehydrated in 100% methanol before storage at −20°C overnight. Larvae were then rehydrated in graded methanol/0.1% PBS-Tween 20 washes, treated with 10 µg/ml proteinase K for 10 min and postfixed with 4% paraformaldehyde for 20 min. Double transgenic *Tg(HGn39d; myo6b:actb1-EGFP)* 8 dpf larvae were simultaneously probed for *col18a1a* (NM_001349195.1) and *sox10* (NM_131875.1) mRNA either 1 day post nerve lesion or in larvae where the nerve was left intact. Fixed *Tg(HGn39d; myo6b:actb1-EGFP)* larvae were also separately probed for *agrn* (NM_001177452.1) and for *lingo3a* (NM_001082854.1) mRNA. Single transgenic *Tg(krt19:col1α2-GFP)* 8 dpf larvae were probed for *col18a1a*. We skipped the proteinase K incubation step for these larvae to better preserve the structure of the ECM. Larvae were incubated with RNA probes overnight at 37°C followed by incubation with corresponding fluorescent hairpin markers overnight at room temperature. To aid in cell counting, a 1:200 dilution of 1 mg/ml DAPI was used in the final wash step before mounting larval trunks in ProLong Diamond Antifade (Invitrogen; P36961) under a 1.5 thickness coverslip. Confocal stacks of individual neuromasts were obtained at 0.200 µm intervals along the sagittal axis. Double positive *col18a1a*+/*sox10*+ Schwann cells were identified by measuring the intensity of probe signal around single nuclei in the confocal stack. Contrast adjustments, cell counting and three-dimensional distance measurements were performed in Fiji and Imaris.

### Extracellular matrix imaging and puncture

The collagen I epidermal boundary layer in relation to pLL nerve afferent axons was imaged in live 5 dpf *Tg(krt19:col1α2-GFP; hsp70:NTR2.0-2A-mCherry en.Sill1)* larvae. The collagen I epidermal boundary layer in relation to primordium-derived neuromast and interneuromast cells was imaged in live 4 dpf *Tg(krt19:col1α2-GFP; she:H2A-mCherry)* larvae. All confocal stacks were obtained at 0.200 µm intervals along the sagittal axis. Gap area measurements, three-dimensional distance measurements, and orthogonal slicing of the confocal stack was performed in Fiji and Imaris.

Confocal stacks of primary L3 or L4 neuromasts were obtained in *Tg(krt19:col1α2-GFP; hsp70:NTR2.0-2A-mCherry en.Sill1)* larvae pre-nerve lesion at 5 dpf, 1 day post lesion at 6 dpf and 3 days post lesion at 8 dpf. In between image acquisition, larvae were freed from the agarose used

for immobilization and left to recover in blue water. The somite number at which a neuromast was imaged pre-nerve lesion was used as a fiduciary marker for repeat imaging of the same neuromast 1 day and 3 days post lesion. Stacks were viewed in Imaris to assess the number of neuromasts re-innervated through the pre-existing gap in the ECM. The area of the gap and its circularity were also measured from regions of interest manually defined in ImageJ.

A glass microneedle pulled to a diameter of ∼20 µm was mounted on a stereotactic rig to precisely puncture the skin on the lateral side of 5 dpf *Tg(krt19:col1α2-GFP; hsp70:NTR2.0-2A-mCherry en.Sill1)* larvae somewhere along the horizontal myoseptum, directly after nerve lesion. Larvae were then left to recover in blue water for 3 days before imaging. The artificial gap created by needle puncture was located by scanning along the length of the regenerated pLL nerve for a large, jagged-edged hole in the collagen I matrix. Because the larvae were poked at a random location along the horizontal myoseptum, the artificially made gap occurred either on an open region of the nerve with no neuromasts nearby or on a region of the nerve close to a neuromast. In the latter case, the artificial gap was anterior to, posterior to or encompassed the natural pre-existing gap in the ECM. Larvae in which punctures coincided directly with the natural gap were excluded from analysis. Confocal stacks were viewed in Imaris to assess the extension of axons into the epidermis through artificial gaps. Epidermal extension was defined as complete passage of an axon into the epidermis, and growth beyond the borders of the gap.

### Mutant generation and phenotype scoring

The *col18a1a*^ru703 line was created through CRISPR/Cas9 gene editing of wild-type embryos and selection of a stable germline mutation following standard protocols (Kroll et al., 2021). Potential target sites for gene editing were determined using ChopChop (v3 https://chopchop.cbu.uib.no/). A 14.25 µM ribonucleoprotein complex of sgRNA (IDT) designed against an exon common to all three major isoforms of Col18a1a and high-fidelity Cas9 (IDT; 1081061) was injected into the yolk of single-cell embryos. Adult founders, with germline transmission of a 2 bp deletion resulting in a frameshift and premature stop codon in exon 2 (201, 203 isoform) or exon 3 (202 isoform), were identified by Sanger sequencing of offspring and using the PolyPeak Parser tool (Hill et al., 2014). These adult founders were outcrossed to *Tg(HGN39d; myo6b:actb1-GFP)* adults and F1 heterozygote offspring were raised to sexual maturity. All experiments were performed on F2 larvae with fluorescent lateral line afferent neuron and hair cell labels. Similarly, *agrn*^p168 mutants with a 7 bp deletion/2 bp insertion in the donor splice site of exon 31 (Gribble et al., 2018) were outcrossed to *Tg(HGn39d; myo6b:actb1-GFP)* adults to incorporate a lateral line afferent and hair cell marker. Heterozygotes were raised to sexual maturity and experiments performed on offspring.

Homozygous mutants and nonmutant siblings were probed at 5 dpf for *col18a1a* RNA using standardized HCR protocols and imaged as described in the section on RNA fluorescence *in situ* hybridization. Probe intensity was quantified in ImageJ by manually defining a region of interest around hair cells of the neuromast in maximum projection images that encompassed the imaging planes of the hair cells. Total RNA was isolated from homozygous mutants using a combination of TRIzol (Invitrogen) extraction and column-based purification with the RNeasy Plus Micro Kit (QIAGEN) as previously described (Lan et al., 2009). Total RNA was reverse transcribed using the SuperScript IV Reverse Transcriptase (Invitrogen) and following the manufacturer's instructions. The resulting cDNA library was sequenced with *col18a1a*-specific primers flanking the mutation site and chromatograms were visualized with SnapGene Viewer.

Axon bundle width was measured on maximum projections of confocal stacks taken from *col18a1a* mutants and siblings at the level of the 10th and 15th somite. We did not observe any phenotypic differences between *agrn*^+/p168 and *agrn*^+/+ siblings, or *col18a1a*^ru703/+ and *col18a1a*^+/+ siblings, and thus grouped them together for analysis. For both *agrn*^p168 and *col18a1a*^ru703 mutant larvae and wild-type siblings, the right pLL nerve was lesioned at 5 dpf and larvae left to recover for 3 days in blue water. At 8 dpf, the regenerated pLL nerve was scanned for aberrant axon pathfinding. Aberrant pathfinding was defined as the inappropriate defasciculation of axons at locations where there were no neuromasts or the extension of axons

beyond the terminal neuromast. The genotype of the larvae was unknown during analysis and was then determined through Sanger sequencing of larvae after phenotype scoring. Each experiment was repeated on at least three separate clutches.

## Statistical analysis

Statistical testing for differential gene expression in the bulk RNA sequencing data was performed using the DESeq2 package in R, employing the Wald test with a correction for multiple comparisons using the Benjamini and Hochberg method. Normalized enrichment scores were generated using the fgsea package in R, using the fast gene set enrichment analysis method with a correction for multiple comparisons using the Benjamini and Hochberg method. Additional analysis was performed in GraphPad Prism 10 using an unpaired Student's *t*-test for comparison of quantitative data, a chi-squared test for testing independence between categorical variables, and Fisher's exact test when sample size assumptions for the chi-squared test were not met. Statistical details such as sample size and significance cutoffs are given in the respective figure legends.

## Acknowledgements

We thank Samantha Campbell for her expert care of the fish facility, Gaurav Shrestha for his assistance performing nerve lesions, Agnik Dasgupta for comments on axon-ECM interactions and members of our research group for their comments on the manuscript. We also thank members of the Flow Cytometry Resource Center, Genomics Resource Center and Bioinformatics Resource Center at The Rockefeller University for their support and guidance. The *Tg(HGn39d)* line was generated through the National BioResource Project (NBRP, Japan).

## Competing interests

The authors declare no competing or financial interests.

## Author contributions

Conceptualization: R.S.R., A.J.H.; Data curation: R.S.R.; Formal analysis: R.S.R.; Funding acquisition: R.S.R., A.J.H.; Investigation: R.S.R., A.J.H.; Methodology: R.S.R.; Project administration: A.J.H.; Resources: A.J.H.; Supervision: A.J.H.; Validation: R.S.R.; Visualization: R.S.R., A.J.H.; Writing – original draft: R.S.R.; Writing – review & editing: R.S.R., A.J.H.

## Funding

R.S.R. was supported by a Medical Scientist Training Program grant from the National Institute of General Medical Sciences of the National Institutes of Health under award number T32GM152349 to the Weill Cornell/Rockefeller/Sloan Kettering Tri-Institutional MD-PhD Program and by a F30 Predoctoral Fellowship from the National Institute on Deafness and Other Communication Disorders of the National Institutes of Health under award number F30DC021350. A.J.H. is an investigator of the Howard Hughes Medical Institute. Open Access funding provided by The Rockefeller University. Deposited in PMC for immediate release.

## Data and resource availability

All raw sequencing data have been deposited in GEO under accession number GSE297706. Quality control and analysis of bulk RNA sequencing data was performed with custom code written by The Rockefeller University Bioinformatics Resource Center and is publicly available on Github (https://github.com/rsroy27/CollagenXVIII_2025_Manuscript). All other relevant data and details of resources can be found within the article and its supplementary information.

## Peer review history

The peer review history is available online at https://journals.biologists.com/dev/lookup/doi/10.1242/dev.205054.reviewer-comments.pdf

## Special Issue

This article is part of the Special Issue 'The Extracellular Environment in Development, Regeneration and Stem Cells', edited by Alex Hughes and Rashmi Priya. See related articles at https://journals.biologists.com/dev/issue/153/16

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
