## [Peer Review File · Development (Cambridge, England)]

Axonal defasciculation is restricted to specific branching points during regeneration of the lateral line nerve in zebrafish

Rohan S. Roy and A. J. Hudspeth

DOI: 10.1242/dev.205054

Editor: Kenneth Poss

Review timeline

Original submission:	23 June 2025
Editorial decision:	4 August 2025
First revision received:	27 October 2025
Editorial decision:	27 November 2025
Second revision received:	30 November 2025
Accepted:	1 December 2025

Original submission

First decision letter

MS ID#: dev.205054

MS TITLE: Axonal defasciculation is restricted to specific branching points during regeneration of the lateral line nerve in zebrafish

AUTHORS: Rohan S. Roy and Albert James Hudspeth

Dear Dr Hudspeth,

I have now received all the referees' reports on the above manuscript, and have reached a decision. The referees' comments are appended below, or you can access them online: please go to: *****

As you will see, the referees express interest in your work, but have some significant criticisms and recommend a substantial revision of your manuscript before we can consider publication. If you are able to revise the manuscript along the lines suggested, which may involve further experiments such as more thorough characterization of mutants, I will be happy to receive a revised version of the manuscript. Your revised paper will be re-reviewed by the original referees, and acceptance of your manuscript will depend on your addressing satisfactorily the reviewers' major concerns. Please also note that Development will normally permit only one round of major revision. If it would be helpful, you are welcome to contact us to discuss your revision in greater detail. Please send us a point-by-point response indicating your plans for addressing the referees' comments, and we will look over this and provide further guidance.

Please attend to all of the reviewers' comments and ensure that you clearly highlight all changes made in the revised manuscript. Please avoid using 'Tracked changes' in Word files as these are lost in PDF conversion. I should be grateful if you would also provide a point-by-point response detailing how you have dealt with the points raised by the reviewers in the 'Response to Reviewers' box. If you do not agree with any of their criticisms or suggestions please explain clearly why this is so.

Reviewer 1

Advance summary and potential significance to field

Regenerating peripheral nerves interact with extracellular signals that influence axon targeting to mediate functional recovery. After nerve transection, the zebrafish posterior lateral line (pLL) nerve reinnervates neuromasts. This process requires fasciculated growth along the pLL and subsequent defasciculation of several axons that innervate individual neuromasts, but the extracellular cues that mediate these distinct processes are unknown. By visualizing the ECM with live imaging, the authors observe that regenerating axons only defasciculate through their original gaps in the ECM. The authors employ bulk RNA sequencing of sorted neuromasts to identify transcripts that change at 1 and 3 days post pLL transection and find that most transcriptional changes that occur at the earlier timepoint are resolved by 3 days when functional reinnervation has occurred. They find that *col18a1a*, which encodes the secreted glycoprotein collagen XVIII, is required to prevent inappropriate axon branching, but is dispensable for axon regrowth and defasciculation as axons grow towards a neuromast. The authors observe via fluorescent in situ that Schwann cells and neuromasts both secrete collagen XVIII, yet they strongly imply throughout the text that *col18a1a*⁺ Schwann cells play a critical role in axon defasciculation towards the neuromast. However, without cell type-specific rescue or knockout, their data do not support this conclusion. Regardless, the RNA sequencing of denervated neuromasts provides a rich dataset and the discovery of a regeneration-specific role of *Col18a1a* to promote appropriate axon targeting, may be of broad interest to the fields of neuromast and axon development and regeneration. Moreover, while future work will be needed to delineate mechanism and significance, this manuscript makes a series of interesting observations including that the ECM gap is maintained after axon injury and regenerating axons preferentially exit there and *col18a1a* is dynamically expressed in Schwann cells after axon injury.

Comments for the author

- 1) The model suggests that a "collagen XVIII-based axon-guidance cue complex attracts defasciculated axon branches across the epidermal basement membrane" (line 31-32) and "We propose that collagen XVIII....scaffolds with axon guidance cues to create a signaling pathway from one side of the epidermal basement membrane to the other" (line 336-338), yet my understanding of the data does not support this conclusion as there was no deficit in *col18a1a*^{-/-} axons reinnervating neuromasts and/or exiting the pLL via the original gap in the ECM. In combination with Fig 7A-B in which the authors report a developmental axon defasciculation defect (tortuosity) in the lateral line in mutants, it seems more accurate to suggest that *Col18a1a* promotes axon fasciculation but does not instruct axons as they cross the basement membrane.
- 2) Likewise, the authors directly suggest a role of *col18a1a* specifically in Schwann cells. For example, "Our results suggest an additional role for *col18a1a*⁺ Schwann cells that restrict defasciculation of axons to specific choice points along the trunk of the fish" (lines 333-334). While Schwann cells may express *col18a1a*, so do neuromasts and possibly other cell types. Without cell-type specific rescue or knockdown, the data in this manuscript do not support such strong statements for the role of *col18a1a* in Schwann cells. I recommend the authors temper the language in their conclusions to be more reflective of the data presented.
- 3) It is interesting that *col18a1a* is downregulated in Schwann cells after injury, which is counterintuitive given the author's model of the role *Col18a1a* plays during regeneration. This observation seems worthy of discussion.
- 4) The Agrin mutant (p168) used is specific to the z isoform typically secreted from neurons. Could the authors clarify which isoforms of agrin were expressed in their RNAsequencing data? Including details regarding the isoform as a caveat would help with interpreting the lack of phenotype in agrin mutants.

Minor:

- 1) Fig 7B is not labeled
- 2) Supp Fig 6 could benefit from a schematic as it was not clear what these data demonstrate and where on the fish was imaged.
- 3) Please clarify whether the axon defasciculation phenotypes shown in fig 7C & D occur medial or lateral to the ECM.

4) While the orientation of images is clearly described in the methods, Fig 4 could benefit from either a schematic or labels denoting medial vs lateral to help orient readers.

Reviewer 2

Advance summary and potential significance to field

In this study, Roy et al. investigate the molecular cues guiding axon pathfinding during peripheral nerve regeneration using the larval zebrafish posterior lateral line as a model. This system is particularly well-suited for such studies due to its innate capacity to reinnervate hair cells in mechanosensory neuromast organs after injury. Through live imaging and pharmacological manipulation, the authors demonstrate that axon regeneration proceeds in two distinct phases: first, axon bundles extend along the horizontal myoseptum; second, follower axons defasciculate from the main bundle at defined points along the trunk. Bulk RNA sequencing of denervated hair cells at 1- and 3-days post-lesion (dpl), compared to sham controls, identifies col18a1a, which encodes the heparan sulfate proteoglycan collagen XVIII, as transiently upregulated at 1 dpl. RNA-FISH shows col18a1a expression in neuromasts both before and after lesion, and strikingly in a single Schwann cell located at the site of axon defasciculation. The authors propose that Col18a1a acts as a guidance cue, facilitating defasciculation at natural, preexisting gaps in the extracellular matrix (ECM), thereby directing axons to their targets. While stable knockout of col18a1a disrupts the precision of axon defasciculation, it does not prevent neuromast reinnervation, indicating that col18a1a is not essential for target recognition but is required for the spatial fidelity of axonal navigation.

The finding that col18a1a expression in a single Schwann cell contributes to axon defasciculation is interesting, as is the observation that regenerating axons appear to follow pre-established ECM gaps used during development. The authors employ a combination of standard approaches - including transcriptomics, pharmacological manipulation, and in vivo imaging - to support their conclusions. Below are some concerns that I believe should be addressed to strengthen the manuscript and advance it towards publication.

In Figure 1C-D, AG1478 is used to disrupt ErbB signaling in Schwann cells. However, the authors do not provide data showing how this treatment affects Schwann cell migration or behavior in their system - these analyses should be included.

The authors report that AG1478 reduces leader axon extension but does not alter axon defasciculation. Does AG1478 impair the presence or function of the col18a1a⁺ Schwann cell positioned at the defasciculation point? If this cell provides a key guidance cue, wouldn't disrupting its migration with AG1478 also lead to aberrant defasciculation?

The selection of 1 and 3 dpl for transcriptomic analysis is based on the assumption that neuromasts are mostly denervated at 1 dpl and reinnervated by 3 dpl. However, Supplemental Figure 1 does not include images of the 3 dpl timepoint, preventing validation of this claim. Please provide representative images at 3 dpl to confirm reinnervation status.

Line 183-184: "Our bulk sequencing data indicate that a transient change in the gene expression of denervated hair cells is in part due to regeneration of lost synapses." Enrichment of synapse gene-ontology sets rather suggests that synaptic genes are modulated in hair cells after injury versus controls. Please clarify.

Are agrn and col18a1a the only HSPGs genes upregulated following injury? Given the reported GO enrichment, one would expect additional HSPG-encoding genes to be upregulated. What criteria were used to select col18a1a for further study? Was it the most strongly upregulated HSPG?

In Figure 3A, a single col18a1a-expressing, sox10⁺ Schwann cell is identified by RNA-FISH, but the image lacks clarity. Inclusion of nuclear and membrane markers is necessary to clearly visualize and localize this specific cell.

Is the single Col18a1a⁺ Schwann cell consistently present at the defasciculation point regardless of lesion location? For example, what happens if the lesion is made more rostrally or closer to the 11th neuromast? Is col18a1a expression maintained at new defasciculation sites, or is its localization context-dependent?

The study compares nerve lesion to a sham injury using 30 seconds of UV LED exposure. Considering the tissue disruption caused by laser ablation, is UV exposure an appropriate sham? What tissue damage (if any) does the sham induce?

Bulk RNA-seq shows a transient increase in col18a1a expression in hair cells at 1 dpl, but this is not evident in RNA-FISH. Additionally, RNA-FISH or in situ hybridization at 3 dpl is needed to support the reported transient upregulation.

Moreover, if col18a1a-expressing Schwann cells guide defasciculation, is their expression activated before or only after reinnervation? Current data suggest Col18a1a is absent at 1 dpl in denervated neuromasts and appears only in reinnervated ones, which complicates the proposed role in guiding initial defasciculation.

Line 249: "In 31 of 32 larvae, we observed regeneration through the same ECM gap through which the axons extended before lesioning." Does this reflect an intrinsic property of regenerating axons, such as their ability to retrace original trajectories, or does it suggest the presence of local cues at the ECM gap that actively guide regrowth?

Axon defasciculation occurs at natural ECM discontinuities, not artificial ones. Are these gaps present under homeostatic conditions, or are they generated/enlarged during regeneration? Do they change in size or morphology after injury?

Validation of col1a1 mutants is missing. Are Col1a1 mRNA/protein missing in these mutants? Do fish develop and mate normally?

The authors state that Col18a1a is not essential for hair cell reinnervation, yet its absence leads to disorganized axon bundles. How does global Col18a1a loss lead to looser fasciculation? Would mislocalized Col18a1a (e.g., via ectopic overexpression in Schwann cells) lead axons to defasciculate aberrantly, even in the absence of ECM gaps?

Artificial ECM gaps are significantly larger than native ones (Fig. 6C). Could this difference explain why axons fail to defasciculate there? The morphological and biochemical properties of natural gaps might be critical for permissive signaling.

The authors note no obvious reinnervation defects in agrn mutants. Please include representative images or quantification to support this claim.

Figure 8 proposes that a Collagen XVIII-guidance cue complex directs axon defasciculation at the neuromast. However, data in Figure 7C-E indicate that neuromast innervation occurs regardless of Collagen XVIII presence (Line 279: "Although innervation of neuromasts appeared grossly normal in homozygous mutants..."). The primary defect observed is aberrant defasciculation, which appears to be mitigated by Collagen XVIII. Please revise the text and/or modify the cartoon in Figure 8 to more accurately reflect that Collagen XVIII influences the precision of defasciculation rather than being required for neuromast innervation per se.

In Figure 7, panel labels need correction, as panel B is missing.

Reviewer 3

Advance summary and potential significance to field

Here, the authors present work that characterizes the role of neuron-extrinsic factors in regulating axon pathfinding and neuromast reinnervation during lateral line nerve regeneration in zebrafish. Bulk RNA sequencing showed acute transcriptional changes in hair cells at 1 dpl and the

authors pursued the role of a secreted HSPG protein, col18a1a for the rest of the study. in-situ hybridization revealed an intriguing expression pattern for col18a1a near gaps in the ECM where defasciculation occurs, suggesting a role in axon defasciculation and reinnervation of neuromasts. Mutant assays did not show a requirement for col18a1a in regeneration and defasciculation, but aberrant axon growth patterns and less compaction in axon bundling were noted.

Strengths of this study include the significance of implicating neuron-extrinsic factors (col18a1a and Schwann cells) in pLL regeneration and defasciculation, thorough characterization of axon defasciculation around ECM gaps and a clever experiment showing preferred defasciculation by natural ECM gaps compared to experimentally created gaps.

The major weakness is the lack of functional data implicating either col18a1a or Schwann cells in defasciculation as detailed below.

Comments for the author

1. col18a1a mutant analysis:

- There is a disconnect between the characterization of defasciculation/col18a1a expression and the phenotype of col18a1 mutants. col18a1a mutant axons can regenerate and defasciculate to reinnervate neuromasts, but loose axon bundles and some aberrant axon growth into inappropriate structures are observed. The authors conclude that col18a1 plays an auxiliary scaffold role that restricts defasciculation to specific sites. If this is case, the expectation would be for col18a1 expression to be all along axon bundles and excluded from the defasciculation site which is not the case.

- It seems the primary phenotype is in axon fasciculation in intact animals, which is also inconsistent with gene expression/Schwann cell localization/ECM gap. Do the Schwann cells along the axon bundle express col18a1?

- Only mutant insets are shown at 3 dpl and thorough analysis and quantifications for col18a1a mutants is lacking. Does aberrant axon growth occur through the same gaps in ECM? How do ECM gaps look in the mutants? How much aberrant growth is there between primary and secondary neuromasts? Authors may consider an additional experiment, similar to that shown in figure 6, evaluating axon preferences in col18a1a mutants.

2. RNAseq analysis. It is unclear whether clustering of the 3 dpl samples is reflective of regeneration versus technical issues. Combined PCA analysis of all samples at 1 and 3 dpl is recommended. Authors should confirm that appropriate batch corrections are used. Additional graphs showing fold change for genes of interests, such as those highlighted in 1F or at least col18a1a, with replicates plotted as individual values would be helpful.

Minor comments

3. Validating bulk sequencing with qPCR using standard regeneration markers along with col18a1a is recommended.

4. Figure 3A shows nice expression and colocalization of col18a1a and sox10 at defasciculation sites in intact animals, but the rest of the in situ after lesion are difficult to see or interpret. The authors quantify col18a1a+ cells but it is unclear what threshold or parameters are used to call a positive cells. Colocalization was also not quantified. Thorough quantification of signal intensity is recommended at specific neuromasts.

Figure 3B - Consider adding an additional circle or arrow indicating missing expression in the Schwann cell on the col18a1a panel. Alternatively, an overlaid multichannel image would nicely visualize the point made in the text.

Figure 3D - A stacked bar chart or a bar chart like 3F might be a more intuitive representation of this data. Are there Schwann cells that are negative for col18a1a? How do these numbers and expression levels look in intact versus injured animals?

First revisionAuthor response to reviewers' comments

****Please refer to the formatted PDF of this Response titled "Response to Reviewers" for the complete response with images****

Dear Dr. Poss,

I am grateful to you and the reviewers for their thoughtful comments on our manuscript, "Axonal defasciculation is restricted to specific branching points during regeneration of the lateral line nerve in zebrafish". I am glad to see the reviewers' interest in the work and am appreciative of their constructive feedback. I have included a point-by-point response below that addresses the reviewers' concerns with direct references to changes made in the main text and figures. All changes made in the revised manuscript are highlighted in yellow. Sadly, the original corresponding author of this work and my thesis advisor, Dr. A. James Hudspeth, passed away in August after a yearlong battle with cancer. In his place, I have assumed the role of corresponding author, and I have been carrying out the recommendations of the reviewers. I appreciate the continued flexibility and guidance of the *Development* editorial team as the laboratory enters a transitional stage. Dr. Hudspeth and I were confident that these results would be of interest to the wider neural regeneration community and would inspire further study by other research groups. I hope that this revised manuscript will be appropriate for publication in *Development*.

Reviewer 1

Regenerating peripheral nerves interact with extracellular signals that influence axon targeting to mediate functional recovery. After nerve transection, the zebrafish posterior lateral line (pLL) nerve reinnervates neuromasts. This process requires fasciculated growth along the pLL and subsequent defasciculation of several axons that innervate individual neuromasts, but the extracellular cues that mediate these distinct processes are unknown. By visualizing the ECM with live imaging, the authors observe that regenerating axons only defasciculate through their original gaps in the ECM. The authors employ bulk RNA sequencing of sorted neuromasts to identify transcripts that change at 1 and 3 days post pLL transection and find that most transcriptional changes that occur at the earlier timepoint are resolved by 3 days when functional reinnervation has occurred. They find that col18a1a, which encodes the secreted glycoprotein collagen XVIII, is required to prevent inappropriate axon branching, but is dispensable for axon regrowth and defasciculation as axons grow towards a neuromast. The authors observe via fluorescent in situ that Schwann cells and neuromasts both secrete collagen XVIII, yet they strongly imply throughout the text that col18a1a+ Schwann cells play a critical role in axon defasciculation towards the neuromast. However, without cell type-specific rescue or knockout, their data do not support this conclusion. Regardless, the RNA sequencing of denervated neuromasts provides a rich dataset and the discovery of a regeneration-specific role of Col18a1a to promote appropriate axon targeting, may be of broad interest to the fields of neuromast and axon development and regeneration. Moreover, while future work will be needed to delineate mechanism and significance, this manuscript makes a series of interesting observations including that the ECM gap is maintained after axon injury and regenerating axons preferentially exit there and col18a1a is dynamically expressed in Schwann cells after axon injury.

Response:

I thank the reviewer for their careful reading of the manuscript and for their assessment of the impact of the results on the neuromast and axon development/regeneration fields. I hope the revisions outlined below will address their concerns and provide further clarification.

Point 1.1:

1) *The model suggests that a "collagen XVIII-based axon-guidance cue complex attracts defasciculated axon branches across the epidermal basement membrane" (line 31-32) and "We propose that collagen XVIII....scaffolds with axon guidance cues to create a signaling pathway from one side of the epidermal basement membrane to the other" (line 336-338), yet my understanding*

of the data does not support this conclusion as there was no deficit in col18a1a^{-/-} axons reinnervating neuromasts and/or exiting the pLL via the original gap in the ECM. In combination with Fig 7A-B in which the authors report a developmental axon defasciculation defect (tortuosity) in the lateral line in mutants, it seems more accurate to suggest that Col18a1a promotes axon fasciculation but does not instruct axons as they cross the basement membrane.

Response:

I appreciate the reviewer's concern about the claims made about the role of *col18a1a* and have adjusted the language in the main text to more accurately reflect the data. I agree that there is no direct evidence of *col18a1a* guiding axons across the ECM gap. Our speculative scaffolding model was a way of reconciling the two main phenotypes of the *col18a1a* mutants: (1) excessive inappropriate defasciculation of axons following nerve lesion and (2) looser fasciculation of the nerve (**Fig. 7C-I**), with the observation of selective expression at only a subset of Schwann cells along the nerve, at the locations of axon branching (**Fig. 3A, Supplemental Fig. S6A-E**). Rather than directly signaling to axon growth cones, we have proposed that Collagen XVIII may be binding to axon guidance cues and incorporating these cues into the basement membrane. Secreted HSPGs, like Collagen XVIII, are known to bind canonical axon guidance cues and modulate their activity in the extracellular space¹⁻³. By binding different guidance cues and patterning them to different locations, this signaling complex could have different effects on axons at different locations along the nerve. This could enable a pro- defasciculation role at the branch points near neuromasts and a pro-fasciculation role elsewhere along normally unbranched portions of the nerve. The scaffolding model can also account for the normal reinnervation of neuromasts in the Collagen XVIII mutants, as the true axon guidance cues would still be present in the lateral line, albeit not precisely patterned. I have changed the language in the text to clarify that *col18a1a* is affecting the precise patterning and refinement of axon navigation, rather than providing direct instruction to cross the basement membrane (**Line 31-32, 114-116, 436-438**). I have also added an additional paragraph to the discussion (**Line 372-379**) to further explain the proposed scaffolding model and have modified the cartoon model (**Fig. 8**) and legend text (**Line 1063-1070**).

Point 1.2:

2) Likewise, the authors directly suggest a role of col18a1a specifically in Schwann cells. For example, "Our results suggest an additional role for col18a1a⁺ Schwann cells that restrict defasciculation of axons to specific choice points along the trunk of the fish" (lines 333-334). While Schwann cells may express col18a1a, so do neuromasts and possibly other cell types. Without cell-type specific rescue or knockdown, the data in this manuscript do not support such strong statements for the role of col18a1a in Schwann cells. I recommend the authors temper the language in their conclusions to be more reflective of the data presented.

Response:

I agree that without cell-type specific rescue and knockdown experiments, statements about the role of *col18a1a* in Schwann cells are only speculative, and I have removed them from the discussion. This claim was made because there is only a subset of Schwann cells, at the locations of axon defasciculation, that are selectively expressing *col18a1a* at a high level. I did not observe *col18a1a* expression in Schwann cells elsewhere along the nerve (**Fig. 3A, Supplemental Fig. S6A-E**), nor did I observe salient expression in any additional cell types other than the ones already noted. This seems to suggest that the *col18a1a* is playing a unique role in specialized Schwann cells at defasciculation points.

Point 1.3:

3) It is interesting that col18a1a is downregulated in Schwann cells after injury, which is counterintuitive given the author's model of the role Col18a1a plays during regeneration. This observation seems worthy of discussion.

Response:

I was also intrigued by the diminished expression of *col18a1a* in Schwann cells directly following

injury and have included further discussion in the manuscript text (Line 353-361). The dynamic expression of *col18a1a* in Schwann cells at the axon defasciculation point appears to follow a separate timeline than the expression in the hair cells of the neuromast. I reasoned that the initial downregulation of *col18a1a* in Schwann cells may be due to the de-differentiation of Schwann cells that occurs during Wallerian degeneration and clearance of axonal debris in the lateral line^{4,5}. It is possible that some additional signal is necessary, such as the passage of a pioneering axon, to again turn on the expression of *col18a1a*, or that Schwann cells would autonomously turn on expression as they re-differentiate. The return in *col18a1a* expression may affect the branching of follower axons, which would explain why *col18a1a* expression is again observed only when a neuromast has been reinnervated. Unfortunately, I lack the temporal resolution with my *in situ* staining to determine the exact timepoint at which *col18a1a* expression reappears in the Schwann cell. I can only narrow it down to between when a pioneering axon passes (Fig. 3C) and when a neuromast is reinnervated by a defasciculated follower axon (Fig. 3D). The dynamic expression observed in Schwann cells is still congruent with the proposed scaffolding model, as Collagen XVIII would still be expressed by both Schwann cell and neuromast at the time of the defasciculation of a follower axon.

Point 1.4

4) The Agrin mutant (p168) used is specific to the z isoform typically secreted from neurons. Could the authors clarify which isoforms of *agrn* were expressed in their RNAsequencing data? Including details regarding the isoform as a caveat would help with interpreting the lack of phenotype in *agrn* mutants.

Response:

I thank the reviewer for bringing this detail to my attention. As I only performed single-end sequencing of bulk RNA, I was unable to determine which specific isoform of *agrn* was expressed by the hair cells. I have included this as a caveat in the discussion (Line 429-431).

Minor

Point 1.5

1) Fig 7B is not labeled

Response:

I have added back the missing label for the panel, which is now Fig. 7D.

Point 1.6

2) Supp Fig 6 could benefit from a schematic as it was not clear what these data demonstrate and where on the fish was imaged.

Response:

I have added additional labels to what is now **Supplemental Fig. S7** and included a maximum projection image of the confocal stack to further clarify what is being shown. The purpose of these images is to contrast the clear, circumscribed gaps observed adjacent to primary neuromasts (**Fig. 4A, Supplemental Video 2**) with the less-defined ‘tunnel’ through the ECM that is observed adjacent to secondary neuromasts. At the entrance of the tunnel, the defasciculated axons are just medial to the collagen I mesh. In the tunnel, the defasciculated axons are embedded within the collagen I mesh. At the end of the tunnel near the neuromast, the defasciculated axons emerge on the lateral side of the collagen I mesh.

**Point 1.7**

3) Please clarify whether the axon defasciculation phenotypes shown in fig 7C & D occur medial or lateral to the ECM.

Response:

In order to localize aberrant axon branching with respect to the ECM, I incorporated the *col18a1a^{tu703}* mutant allele into the *Tg(HGn39d; krt19:col1a2-GFP)* transgenic background. I was unfortunately unable to produce homozygous mutant offspring with complete labeling of the collagen I-GFP mesh. However, I was able to produce homozygous mutants with partial labeling of the epidermal boundary layer and have presented images of aberrant branching following nerve lesion in **Supplemental Fig. S8C**. Even with partial labeling of the collagen I mesh, I was able to localize aberrant branching as crossing over to the lateral side of the ECM. These crossing points were distinct from the native gaps shown in **Fig. 4** and **Fig. 5** that permit entry into the epidermis in wildtype fish. I have added additional text to the manuscript describing this result (**Line 307-310**). This further confirms that *col18a1a* is not required for axons to cross the basement membrane. Rather, *col18a1a* is likely playing an auxiliary role in refining and patterning axon branching to appropriate locations.

**Point 1.8**

4) While the orientation of images is clearly described in the methods, Fig 4 could benefit from either a schematic or labels denoting medial vs lateral to help orient readers.

Response:

I have added labels denoting medial and lateral to the coronal images shown in Fig. 4.

Reviewer 2

In this study, Roy *et al.* investigate the molecular cues guiding axon pathfinding during peripheral nerve regeneration using the larval zebrafish posterior lateral line as a model. This system is particularly well-suited for such studies due to its innate capacity to reinnervate hair cells in mechanosensory neuromast organs after injury. Through live imaging and pharmacological manipulation, the authors demonstrate that axon regeneration proceeds in two distinct phases: first, axon bundles extend along the horizontal myoseptum; second, follower axons defasciculate from the main bundle at defined points along the trunk. Bulk RNA sequencing of denervated hair cells at 1- and 3- days post-lesion (dpl), compared to sham controls, identifies *col18a1a*, which encodes the heparan sulfate proteoglycan collagen XVIII, as transiently upregulated at 1 dpl. RNA-FISH shows *col18a1a* expression in neuromasts both before and after lesion, and strikingly in a single Schwann cell located at the site of axon defasciculation. The authors propose that *Col18a1a* acts as a guidance cue, facilitating defasciculation at natural, preexisting gaps in the extracellular matrix (ECM), thereby directing axons to their targets. While stable knockout of *col18a1a* disrupts the precision of axon defasciculation, it does not prevent neuromast reinnervation, indicating that *col18a1a* is not essential for target recognition but is required for the spatial fidelity of axonal navigation.

The finding that *col18a1a* expression in a single Schwann cell contributes to axon defasciculation is interesting, as is the observation that regenerating axons appear to follow pre-established ECM gaps used during development. The authors employ a combination of standard approaches - including transcriptomics, pharmacological manipulation, and *in vivo* imaging - to support their conclusions. Below are some concerns that I believe should be addressed to strengthen the manuscript and advance it towards publication.

Response:

I thank the reviewer for their interpretation of the manuscript and their interest in the main findings. I particularly like how the reviewer summarized, “*col18a1a* is not essential for target recognition but is required for the spatial fidelity of axonal navigation”. I hope I have satisfactorily addressed their concerns below.

Point 2.1

In Figure 1C-D, AG1478 is used to disrupt ErbB signaling in Schwann cells. However, the authors do not provide data showing how this treatment affects Schwann cell migration or behavior in their system - these analyses should be included.

Response:

Although I have not rigorously tested the effects of AG1478 on the lateral line myself, I have added citations and descriptions of its documented effects on Schwann cells and ErbB signaling in the main text (Line 138-143). The effect of AG1478 is dependent on when it is administered during development. When given prior to 58 hpf, AG1478 partially or completely blocks the migration of Schwann cells along the posterior lateral line nerve.

I administered AG1478 at 4 dpf, well after Schwann cells have completely migrated to their final positions along the nerve. At this timepoint, AG1478 does not affect the existing placement of Schwann cells along the nerve^{4,6}. It does halt the continued proliferation and replacement of Schwann cells along the nerve⁶. I have added additional clarification in the main text that AG1478 was administered post-Schwann cell migration (Line 143-144). I do not believe that there is anything unique about my experimental setup that would change how AG1478 would affect lateral line Schwann cells compared to earlier studies.

Point 2.2

*The authors report that AG1478 reduces leader axon extension but does not alter axon defasciculation. Does AG1478 impair the presence or function of the *col18a1a*⁺ Schwann cell positioned at the defasciculation point? If this cell provides a key guidance cue, wouldn't disrupting its migration with AG1478 also lead to aberrant defasciculation?*

Response:

I have clarified in the main text that AG1478 was administered after Schwann cell migration had completed (Line 143-144). I do not expect that AG1478 administered at 4 dpf would impair the presence or function of the *col18a1a*⁺ Schwann cells that exist at defasciculation points. At this advanced timepoint, my goal was to disrupt ErbB/neuregulin signaling between existing Schwann cells and regenerating axons. Previous work has shown that early administration of AG1478 prior to 2 dpf, which blocks the initial migration of Schwann cells along the nerve, does impair axon pathfinding and causes aberrant defasciculation⁴.

Point 2.3

The selection of 1 and 3 dpl for transcriptomic analysis is based on the assumption that neuromasts are mostly denervated at 1 dpl and reinnervated by 3 dpl. However, Supplemental Figure 1 does not include images of the 3 dpl timepoint, preventing validation of this claim. Please provide representative images at 3 dpl to confirm reinnervation status.

Response:

I have remade **Supplemental Fig. S1** to track regeneration of the nerve 1 day post lesion and 3 days post lesion. By three days post lesion, the pioneering axons of the nerve have regenerated to the tail end of the fish, and all the neuromasts have been reinnervated. Furthermore, the images in **Fig. 5A**, **6A-C**, and **7F-H** were taken at 3 days post lesion, and also show the reinnervation of neuromasts at various locations along the nerve. Prior studies also demonstrate that the lateral line, axotomized at a similar location as my experiments, will regenerate and reinnervate

neuromasts by two to three days post lesion^{4,5,7}.

Point 2.4

Line 183-184: "Our bulk sequencing data indicate that a transient change in the gene expression of denervated hair cells is in part due to regeneration of lost synapses." Enrichment of synapse gene-ontology sets rather suggests that synaptic genes are modulated in hair cells after injury versus controls. Please clarify.

Response:

I agree that the original statement is only speculation based on the enrichment of synapse gene-ontology sets in hair cells following nerve lesion. I thank the reviewer for their rewording of the statement and have substituted it into the main text (Line 194-196).

Point 2.5

Are agrn and col18a1a the only HSPGs genes upregulated following injury? Given the reported GO enrichment, one would expect additional HSPG-encoding genes to be upregulated. What criteria were used to select col18a1a for further study? Was it the most strongly upregulated HSPG?

Response:

There are only three known secreted HSPGs that incorporate into the extracellular matrix: agrin, collagen XVIII, and perlecan⁸. Out of these three, only *agn* and *col18a1a* were significantly upregulated following nerve lesion. There are additional HSPGs that are membrane-bound on epithelial cells, such as glypicans and syndecans. The genes coding for these membrane-bound HSPGs were not differentially expressed following nerve lesion (Fig. 2F). The 'heparan sulfate proteoglycan biosynthetic process' GO set that was enriched comprises genes for enzymes that

build and modify the heparan sulfate side chain, such as glucuronyltransferases, glycosyltransferases, epimerases, and sulfotransferases. This GO set does not include the genes that code for the HSPG core proteins. Therefore, it is not expected that the enrichment of this gene set would be coupled to the upregulation of other HSPG-encoding genes.

The *agr*n and *col18a1a* genes were upregulated to the same degree following nerve lesion (Fig. 2F) and I initially pursued both for further study. I performed *in situ* against both *col18a1a* (Fig. 3A) and *agr*n (Supplemental Fig. S5A). I also obtained an *agr*n mutant and generated my own *col18a1a* mutant to perform nerve regeneration experiments. I did not see any overt reinnervation defects in *agr*n mutants (Supplementary Fig. S5B) and so I chose to further characterize *col18a1a*. Upon performing *in situ* against *col18a1a* RNA, I also observed localized staining at the locations of axon defasciculation, which further prompted me to focus on *col18a1a*.

Point 2.6

In Figure 3A, a single col18a1a-expressing, sox10+ Schwann cell is identified by RNA- FISH, but the image lacks clarity. Inclusion of nuclear and membrane markers is necessary to clearly visualize and localize this specific cell.

Response:

I have revised the images shown in Fig. 3A to better highlight the labeling of a single *col18a1a*⁺/*sox10*⁺ Schwann cell. Unfortunately, I did not have a Schwann cell membrane marker, but I did use DAPI as a nuclear marker for all of my *in situ*. Because there is a sparseness of *col18a1a*⁺ Schwann cells along the nerve – only one to three cells at the locations of axonal defasciculation – I did not need a cell membrane marker to count double positive cells. By inspecting confocal stacks in three-dimensions, I was able to assess *sox10* and *col18a1a* probe signal around individual nuclei.

I have modified Fig. 3A to show the double positive staining around a single nucleus at the axonal defasciculation point. The merged image on the left is a maximum projection of the entire confocal stack. The color channels on the right are from the same individual slice in the confocal stack. The double positive cell is outlined in red.

I have also provided a more detailed quantification of probe signal in a *col18a1a*⁺ Schwann cell compared to a *col18a1a*⁻ Schwann cell in Supplemental Fig. S6A-E. I hope that these images will further clarify how I identified and counted cells.

Point 2.7

Is the single Col18a1a⁺ Schwann cell consistently present at the defasciculation point regardless of lesion location? For example, what happens if the lesion is made more rostrally or closer to the 11th neuromast? Is col18a1a expression maintained at new defasciculation sites, or is its localization context-dependent?

Response:

I observed one to three *col18a1a*⁺ Schwann cells at almost every neuromast along the fish during normal development, at the locations of axonal defasciculation (**Fig. 3F**). I do not believe that the location of nerve lesion would have any effect on where *col18a1a* is expressed, because the locations of axonal defasciculation do not change based on where the nerve is lesioned.

The reviewer might also be referring to the disappearance and reemergence of *col18a1a* probe signal following nerve lesion. I speculate in the discussion (**Line 353-361**) that this pattern may be due to the dedifferentiation and redifferentiation of Schwann cells that matches Wallerian degeneration and axon regeneration. In this case, I would expect to see the changing expression of *col18a1a* in Schwann cells posterior to the nerve lesion site, where Wallerian degeneration takes place, rather than anterior to the lesion site, where the nerve remains intact.

Point 2.8

The study compares nerve lesion to a sham injury using 30 seconds of UV LED exposure. Considering the tissue disruption caused by laser ablation, is UV exposure an appropriate sham? What tissue damage (if any) does the sham induce?

Response:

I believe that the mounting procedure was the predominant source of tissue damage that needed to be controlled. I had to mount and unmount each larva twice in order to lesion both the right and left nerve. This mounting procedure caused superficial tissue damage along the entire body in some larvae. I controlled for this by also mounting and unmounting age-matched sibling larvae twice, so that I could expose each side to a UV light. This mounting procedure was identical to the one used for laser lesioning and thus was assumed to produce the same level of body-wide superficial tissue damage.

I do not believe that the laser ablation itself caused tissue disruption beyond the superficial surface of the fourth somite, where the nerve was lesioned. The ablation was confined to a rectangular region that was approximately 20 μm x 10 μm in area. I only observed ablation of axons at the lesion site, and did not observe any changes to the neuromasts located posterior to the lesion site, several body somites away.

The mounting procedure, exposure to imaging lasers for the localization of the nerve, and ablation also likely induced a stress response in the larvae. During nerve ablation, the larvae was exposed to laser light for approximately 30 seconds. I believe that the sham procedure, which involved the same mounting procedure and exposure to a UV light for 30 seconds, induced a similar stress response.

Point 2.9

Bulk RNA-seq shows a transient increase in col18a1a expression in hair cells at 1 dpl, but this is not evident in RNA-FISH. Additionally, RNA-FISH or in situ hybridization at 3 dpl is needed to support the reported transient upregulation.

Response:

I agree with the reviewer that this is a limitation of the study. I was unable to test differences in RNA-FISH staining at different timepoints following nerve lesion due to the variation in signal between different larvae. In my hands, the hybridization chain reaction (HCR) protocol for

performing RNA-FISH was a powerful tool for localizing the expression of *col18a1a* in space and could be used to determine large changes in gene expression, but was not precise enough to reliably quantify relatively small changes to highly expressed genes.

Bulk RNA sequencing is a more sensitive tool than RNA-FISH in detecting differences in gene expression between similar sample types. The bulk sequencing data indicates a statistically significant 0.741 log₂ fold change of *col18a1a* transcripts in hair cells 1 dpl compared to controls, and no statistically significant change in hair cells 3 dpl compared to controls. I have been able to confirm expression of *col18a1a* in the innervated, denervated, and reinnervated neuromasts (**Fig. 3B-D, dotted blue outline**). However, I have been unable to detect significant *col18a1a* RNA probe intensity changes in neuromasts across larvae, fixed at different timepoints. I have noted this limitation in the manuscript text (**Line number 219-222**). I believe that the additional experiments on *col18a1a* mutants provide functional validation of the importance of this gene following lateral line nerve lesion.

Point 2.10

Moreover, if col18a1a-expressing Schwann cells guide defasciculation, is their expression activated before or only after reinnervation? Current data suggest Col18a1a is absent at 1 dpl in denervated neuromasts and appears only in reinnervated ones, which complicates the proposed role in guiding initial defasciculation.

Response:

I speculate that the loss of *col18a1a* expression in the Schwann cell following nerve lesion is a result of the Schwann cell de-differentiating. I interpret the reemergence of *col18a1a* expression in a subset of Schwann cells, adjacent to recently reinnervated neuromasts, as evidence of a role in patterning axon branching to appropriate locations. I have included additional text commenting on this (**Line 353-361**).

I have also provided additional images to **Fig. 3C** to suggest that the reemergence of *col18a1a* expression in the Schwann cells is somewhere between the passage of a pioneering axon and the reinnervation of a neuromast. The time between axonal defasciculation and neuromast innervation is very short, only one to two hours (**Supplemental Video 1**). Unfortunately, I do not have the temporal resolution in the *in situs* to capture the moment a follower axons defasciculates from the regenerating nerve to reinnervate a neuromast. My images only capture a defasciculated axon that has already reached the neuromast.

I have interpreted the images in **Fig. 3B-D** to mean *col18a1a* expression in the Schwann cell is turning on prior to axon defasciculation. This would explain the staining pattern I observed, in which I could detect *col18a1a*⁺ Schwann cells adjacent to a recently reinnervated neuromast. If *col18a1a* was turned on after neuromast reinnervation, I reasoned that I would have counted more reinnervated neuromasts with a lack of adjacent *col18a1a*⁺ Schwann cells.

Point 2.11

Line 249: "In 31 of 32 larvae, we observed regeneration through the same ECM gap through which the axons extended before lesioning." Does this reflect an intrinsic property of regenerating axons, such as their ability to retrace original trajectories, or does it suggest the presence of local cues at the ECM gap that actively guide regrowth?

Response:

I interpret this result as evidence that there are local cues at the natural ECM gap that are actively guiding regrowth. Previous studies have shown that an individual cut axon is more likely to reinnervate a different neuromast than from before it was cut^{4,9}. This suggests that there is not an intrinsic memory hardwired into any one axon to retrace the exact same trajectory it had during development.

Because all the lateral line axons in the experiment presented in **Fig. 5** are labeled with GFP, it is impossible for me to tell if the exact same axons that passed through an ECM gap pre-nerve lesion are the same as the ones to pass through it again post-nerve lesion. However, I can conclude that axons overwhelmingly pass through the same ECM gap that axons pass through before nerve lesion, rather than through an artificially made gap, suggesting local guidance cues present at the native ECM gap.

Point 2.12

Axon defasciculation occurs at natural ECM discontinuities, not artificial ones. Are these gaps present under homeostatic conditions, or are they generated/enlarged during regeneration? Do they change in size or morphology after injury?

Response:

These gaps are present under homeostatic conditions, as shown in **Fig. 4A**, which shows a gap in a wildtype 5 dpf larva. These gaps are necessary for the passage of defasciculated axons from the nerve, which runs in the subepidermal space, to the neuromast, which resides in the epidermis. These gaps are not changed following nerve lesion, and regenerating axons passed through the same gap rather than going through a newly created one in 31 out of 32 larvae imaged. I have provided further quantification of the gap area and circularity from before lesion, 1 day post lesion, and 3 days post lesion in **Fig. 5C-D**. There is no significant difference in these parameters following injury.

Point 2.13

*Validation of *cola1a1* mutants is missing. Are *Col1a1* mRNA/protein missing in these mutants? Do fish develop and mate normally?*

Response:

I have performed additional validation of the *col18a1a* mutants and have revised **Supplemental Fig. S8** and **Fig. 7** with the results. I collected the total RNA from *col18a1a^{ru703/ru703}* mutants, reverse transcribed it into a cDNA library and sequenced the *col18a1a* cDNA (Materials and Methods, Line 629-639). The chromatogram of this sequencing is shown in **Supplemental Fig. S8B** and shows that the deletion of 2 base pairs from the genomic DNA of mutant larva is also carried into the mRNA. This frameshift results in a premature stop codon downstream of the two base pair deletion that is also carried into the mRNA. I have also performed RNA-FISH against *col18a1a* RNA in *col18a1a^{ru703/ru703}* mutants and compared probe intensity in neuromast hair cells to wildtype fish. I observed a significant decrease in probe intensity in the mutants compared to wildtype larvae, suggesting nonsense mediated decay of the mutated *col18a1a* transcripts (**Fig. 7A,B**; Line 296-299).

I have been unable to find a suitable Collagen XVIII antibody that functions in the zebrafish lateral line and thus have been unable to verify at the protein level any changes in Collagen XVIII in the *col18a1a*^{ru703/ru703}. However, it can be reasonably inferred from the mutated RNA, nonsense-mediated decay of transcripts, and altered nerve phenotype that the protein is likely perturbed in the mutants.

I had previously only raised fish that were heterozygous for the *col18a1a*^{ru703} mutant allele so that I could directly compare mutant and non-mutant sibling offspring. The heterozygous fish developed and mated normally. Since receiving this round of reviewer comments, I attempted to raise homozygous mutants to adulthood. These homozygous mutants are now 2 months old, proving they are viable beyond larval stages, and do not have any overt physical defects or changes to swimming behavior. These fish are not yet sexually mature, so I am unable to determine if they mate normally. While this piece of information would further characterize our mutants, it would not affect the main results of the manuscript.

Point 2.14

The authors state that Col18a1a is not essential for hair cell reinnervation, yet its absence leads to disorganized axon bundles. How does global Col18a1a loss lead to looser fasciculation? Would mislocalized Col18a1a (e.g., via ectopic overexpression in Schwann cells) lead axons to defasciculate aberrantly, even in the absence of ECM gaps?

Response:

I have added additional commentary in the discussion (**Line Number 372-379**) on the altered nerve shape and axon bundling in global *col18a1a*^{ru703/ru703} mutants. I believe that *col18a1a* is generally having an effect on axon bundling and navigation, but the direction of this effect, whether it is pro-bundling or pro-branching, is context dependent.

The proposed scaffolding model is one way of reconciling the selective expression pattern of *col18a1a* at only branching points of the nerve with looser fasciculation in the entire nerve. In this version of the model, Collagen XVIII is presumably secreted by a subset of Schwann cells and is embedded in the basement membrane that spans the entire length of the nerve. The effects of Collagen XVIII would be varied depending on the presence of different complexed guidance cues. It is possible that Collagen XVIII has a pro-bundling effect along the normally fasciculated portions of the nerve, and a pro-branching effect at defasciculation sites.

I would speculate that mislocalized Collagen XVIII through ectopic overexpression in other Schwann cells would adversely affect axon bundling. However, the exact effect would be dependent on where along the nerve it is expressed and what other signaling molecules are present. While I think this is a valuable experiment for further mechanistic characterization of Collagen XVIII, I think it falls outside the immediate scope of this manuscript, which only seeks to

implicate the importance of the *col18a1a* gene in axonal navigation.

Point 2.15

Artificial ECM gaps are significantly larger than native ones (Fig. 6C). Could this difference explain why axons fail to defasciculate there? The morphological and biochemical properties of natural gaps might be critical for permissive signaling.

Response:

I appreciate the reviewer pointing out this distinction between the artificial ECM gaps and the native ones. I have ruled out that a relatively large, crude puncture through the epidermal boundary layer is sufficient for regenerating axons to cross into the epidermis. I agree that it is plausible that both the physical properties and biochemical properties of the smaller native gap may be contributing to selective defasciculation. I have modified the language in the discussion (Line 398-399) to emphasize this.

Point 2.16

The authors note no obvious reinnervation defects in agrn mutants. Please include representative images or quantification to support this claim.

Response:

Quantification for this claim is presented in **Supplemental Fig. S5B**. I did not see any significant difference in aberrant axon pathfinding in *agn^{p168/p168}* larvae following nerve lesion compared to nonmutant siblings.

Point 2.17

Figure 8 proposes that a Collagen XVIII-guidance cue complex directs axon defasciculation at the neuromast. However, data in Figure 7C-E indicate that neuromast innervation occurs regardless of Collagen XVIII presence (Line 279: "Although innervation of neuromasts appeared grossly normal in homozygous mutants..."). The primary defect observed is aberrant defasciculation, which appears to be mitigated by Collagen XVIII. Please revise the text and/or modify the cartoon in Figure 8 to more accurately reflect that Collagen XVIII influences the precision of defasciculation rather than being required for neuromast innervation per se.

Response:

I have changed the language in the text to clarify that *col18a1a* is affecting the precise patterning and refinement of axon navigation, rather than providing direct instruction to cross the basement membrane for neuromast innervation (Line 31-32, 114-116, 436-438). I have also added an additional paragraph to the discussion (Line 372-379) to further explain the proposed scaffolding model and have modified the cartoon model (Fig. 8) and legend text (Line 1063-1070)

to propose that the exact effects of collagen XVIII would be dependent on the guidance cues that it localizes to different portions of the basement membrane.

Point 2.18

In Figure 7, panel labels need correction, as panel B is missing.

Response:

I have added back the missing panel label to what is now **Fig. 7D**.

Reviewer 3

Here, the authors present work that characterizes the role of neuron-extrinsic factors in regulating axon pathfinding and neuromast reinnervation during lateral line nerve regeneration in zebrafish. Bulk RNA sequencing showed acute transcriptional changes in hair cells at 1 dpl and the authors pursued the role of a secreted HSPG protein, col18a1a for the rest of the study. in-situ hybridization revealed an intriguing expression pattern for col18a1a near gaps in the ECM where defasciculation occurs, suggesting a role in axon defasciculation and reinnervation of neuromasts. Mutant assays did not show a requirement for col18a1a in regeneration and defasciculation, but aberrant axon growth patterns and less compaction in axon bundling were noted.

Strengths of this study include the significance of implicating neuron-extrinsic factors (col18a1a and Schwann cells) in pLL regeneration and defasciculation, thorough characterization of axon defasciculation around ECM gaps and a clever experiment showing preferred defasciculation by natural ECM gaps compared to experimentally created gaps.

The major weakness is the lack of functional data implicating either col18a1a or Schwann cells in defasciculation as detailed below.

Response:

I thank the reviewer for their comments and constructive feedback. I am pleased to hear the strengths of the study, and I hope to address the major weaknesses below and in the revised manuscript text.

Point 3.1.1

col18a1a mutant analysis:

-There is a disconnect between the characterization of defasciculation/col18a1a expression and the phenotype of col18a1 mutants. col18a1a mutant axons can regenerate and defasciculate to reinnervate neuromasts, but loose axon bundles and some aberrant axon growth into inappropriate structures are observed. The authors conclude that col18a1 plays an auxiliary scaffold role that restricts defasciculation to specific sites. If this is case, the expectation would be for col18a1 expression to be all along axon bundles and excluded from the defasciculation site which is not the case.

Response:

I agree with the reviewer that there is a discrepancy between the spatial expression of *col18a1a* and the phenotype of the *col18a1a* mutants. I found that *col18a1a* is selectively expressed by a subset of Schwann cells at the locations of axon defasciculation towards neuromasts. This expression pattern would suggest that Collagen XVIII is involved with axon branching at these locations. However, *col18a1a^{ru703/ru703}* mutants have looser axon bundles throughout the nerve and excessive branching at inappropriate locations, suggesting that Collagen XVIII is having a pro-fasciculation effect along the nerve. The proposed scaffolding model was one way of resolving this discrepancy. While *col18a1a* seems to be expressed only at specific locations along the nerve, the secreted Collagen XVIII protein can still embed itself within the basement membrane that spans

the entire nerve and have nerve-wide effects. While Collagen XVIII is likely not directly signaling to axon growth cones, it could be involved in the patterning of different guidance cues in the extracellular matrix. Depending on what cues it is complexed to, Collagen XVIII could have different pro-bundling or pro-branching effects that vary in space and time along the nerve.

I concede that with the current study, I am unable to determine the exact context- dependent effects of the active protein on the nerve (**Line 372-379**). Although I have shown in **Fig. 5** and **Fig. 6** that axon defasciculation is restricted to specific locations during nerve regeneration, I cannot claim that *col18a1a* is directly responsible for this, and I have removed such claims from the text. Instead of restricting axon defasciculation, I have modified the text to clarify that *col18a1a* is necessary for the precision of axon pathfinding, and that its effects on the nerve can be varied.

Point 3.1.2

-It seems the primary phenotype is in axon fasciculation in intact animals, which is also inconsistent with gene expression/Schwann cell localization/ECM gap. Do the Schwann cells along the axon bundle express col18a1?

Response:

I only observe *col18a1a* expression in a small subset of Schwann cells that lie at the point of axonal defasciculation. The majority of *sox10*⁺ Schwann cells that ensheath the linear, unbranched portions of the nerve, did not stain with a *col18a1a* probe. This can be seen in **Fig. 3B-D**, and I have added an additional supplemental figure (**Supplemental Fig. S6**) and text (**Line 212-214**) to further emphasize this.

Given that *col18a1a* expression is restricted to specific locations along the nerve, it is interesting that there were nerve-wide defects in axon bundling and tortuosity. It is possible that the secreted Collagen XVIII protein is embedded within the extracellular matrix throughout the length of the nerve. The localization of the protein along the nerve and in the extracellular space would be a valuable goal for future studies.

Point 3.1.3

-Only mutant insets are shown at 3 dpl and thorough analysis and quantifications for col18a1a mutants is lacking. Does aberrant axon growth occur through the same gaps in ECM? How do ECM gaps look in the mutants? How much aberrant growth is there between primary and secondary neuromasts? Authors may consider an additional experiment, similar to that shown in figure 6, evaluating axon preferences in col18a1a mutants.

Response:

The aberrant branching does not occur through the same gaps that normal defasciculation towards neuromasts occurs through, and I have clarified this in the text (**Line 307-310**). The aberrant axon branching is in addition to the expected defasciculation of axons towards neuromasts, and these inappropriate branching locations occur at locations along the nerve between neuromasts.

I have performed additional experiments and analysis of the *col18a1a*^{ru703/ru703} mutants to address the reviewer's questions to the best of my ability in a reasonable timeframe. In order to localize the aberrant branching with respect to the ECM, I incorporated the *col18a1a*^{ru703} mutant allele into the *Tg(HGn39d; krt19:col1a2-GFP)* transgenic background. Unfortunately, as a result of this outcrossing, I was unable to produce homozygous mutant offspring with complete labeling of the collagen I-GFP mesh.

However, I was able to produce homozygous mutants with partial labeling of the epidermal boundary layer, and have presented images of aberrant branching following nerve lesion in **Supplemental Fig. S8C**. Even with partial labeling of the collagen I mesh, I was able to localize aberrant branching as crossing over to the lateral side of the ECM. These crossing points were distinct from the native gaps shown in **Fig. 4** and **Fig. 5** that permit entry into the epidermis in

wildtype fish.

I have performed additional quantification of *col18a1a* transcript levels between mutants and non-mutant siblings (Fig. 7A,B). There is a significant decrease in RNA-FISH probe intensity in mutant neuromast hair cells compared to non-mutant siblings, suggesting nonsense-mediated decay of the mutated transcripts. I also reverse transcribed mutant RNA and sequenced the resulting cDNA to confirm the two base pair deletion in mutants and the creation of a premature stop codon in mRNA transcripts (Supplemental Figure S8B).

Because I could not observe complete labeling of the collagen I-GFP mesh in homozygous mutant larvae, I could not determine if the natural ECM gaps that permit normal defasciculation towards neuromasts was changed. The incomplete labeling of the Collagen I epidermal boundary layer also prevented me from making artificial punctures in mutant larvae and assessing axon pathfinding. It is unknown to me why the transgenic labeling of the extracellular matrix was affected in the *col18a1a* mutants. It is possible that disrupting *col18a1a* may also affect Collagen I deposition. However, I would expect a more severe phenotype in mutant larvae if this were the case. It is also possible the collagen I-GFP transgene and the mutated *col18a1a* allele occupy a similar region of the genome, and expression was lost as the result of outcrossing. Further investigation into this issue would require additional time and resources, and while interesting, I do not believe it would substantially change the conclusions of the present study.

Point 3.2

RNAseq analysis. It is unclear whether clustering of the 3 dpl samples is reflective of regeneration versus technical issues. Combined PCA analysis of all samples at 1 and 3 dpl is recommended. Authors should confirm that appropriate batch corrections are used. Additional graphs showing fold change for genes of interests, such as those highlighted in 1F or at least col18a1a, with replicates plotted as individual values would be helpful.

Response:

I have performed a combined PCA analysis of all sequenced samples, using the limma package in R for batch correction. The code for the combined analysis has been uploaded to the github page associated with the sequencing analysis for this manuscript (https://github.com/rsroy27/CollagenXVIII_2025_Manuscript).

I have plotted the first two principal components for each sample from this combined analysis and included the plot in Supplemental Fig. S3E. Even with a combined analysis of all samples, there is still greater separation between the 1 dpl samples and their age-matched controls than between 3 dpl samples and their age-matched controls. This has not changed my initial interpretation of the sequencing results, which suggests that transcriptional changes seen 1 day post lesion are lessened by 3 days post lesion, at which time most of the neuromasts have been reinnervated (Supplemental Fig. S1). Biologically, this makes intuitive sense. After hair cells are reinnervated by the nerve, they are indistinguishable from normally innervated hair cells. I would not expect there to be permanent changes to the hair cell following nerve lesion.

Since the combined analysis does not change the initial interpretation of the separated analysis, I have kept the individual PCA plots in **Fig. 2B,D**. I believe that the mixed clustering of hair cells 3 dpl with age-matched controls is reflective of regeneration and not technical issues. These samples were collected from tank siblings, sequenced together on the same chip, and analyzed together. I am unaware of any technical issues that may have influenced the analysis. Although I have performed the combined PCA analysis with batch corrections, I caution that this approach may introduce more technical artifacts than direct comparison of hair cells 1 dpl and 3 dpl to their respective age-matched controls. Hair cell samples 1 dpl were sequenced on the same chip as age-matched controls on an Illumina NextSeq500 and thus were directly compared to one another. Due to changes in our genomics core facility, hair cell samples 3 dpl were sequenced on the same chip as age-matched controls on an Illumina NextSeq2000 and analyzed separately from the 1 dpl samples.

I have plotted the individual normalized transcript counts from each replicate for *col18a1a*, *agrn*, *gpc4*, *gpc1a*, *sdc2*, and *sdc4* in **Supplemental Fig. S3C,D**. There is a significant increase in *col18a1a* and *agrn* transcript counts in hair cells 1 dpl compared to age-matched controls. There is no significant difference in the transcripts in hair cells 3 dpl compared to age-matched controls. The individual transcript counts for every gene that was sequenced is available in the excel files uploaded as supplementary material (**Supplementary File 1 and File 2**).

Minor

Point 3.3

Validating bulk sequencing with qPCR using standard regeneration markers along with col18a1a is recommended.

Response:

Although I would like to further validate the changes observed in the bulk sequencing data with qPCR, I could not perform such experiments in a reasonable timeframe for these revisions. There are no published and validated qPCR primers for quantifying *col18a1a* expression in the zebrafish. Performing qPCR would require additional nerve lesioning and additional FACS sessions to generate hair cell-specific samples. Generating the appropriate number of hair cell samples would take several months. Furthermore, qPCR experiments are prone to technical errors and would require their own set of vigorous controls and multiple rounds of optimization. I have alternatively validated the spatial expression of *col18a1a* in neuromast hair cells through RNA-FISH (Fig. 3) and its functional importance through experimentation with *col18a1a* mutant fish (Fig. 7).

Point 3.4.1

Figure 3A shows nice expression and colocalization of col18a1a and sox10 at defasciculation sites in intact animals, but the rest of the in situ after lesion are difficult to see or interpret. The authors quantify col18a1a+ cells but it is unclear what threshold or parameters are used to call a positive cells. Colocalization was also not quantified. Thorough quantification of signal intensity is recommended at specific neuromasts.

Response:

Because of the sparseness of *col18a1a* expression in Schwann cells along the nerve, I was able to use the nuclear marker DAPI to call individual positive cells. I have added a statement in the Materials and Methods section to clarify this (Line 576-578). I have also modified Fig. 3A to include a single imaging plane capturing signals from DAPI, *col18a1a* RNA probe, and *sox10* RNA probe in a single cell at the point of axonal defasciculation. In most cases, it was obvious when to call a double-positive cell by also using the DAPI marker and scrolling through the confocal stack. I have provided quantification of this process in Supplemental Fig. S6D,E. In this figure, I have measured the intensity of both *sox10* and *col18a1a* probe signals around the nuclei of two different Schwann cells along the nerve. Both Schwann cells have relatively high *sox10* labeling. In the Schwann cell adjacent to axonal defasciculation, there is high *col18a1a* signal, and in the Schwann cell located on an unbranched segment of the nerve, there is a lower *col18a1a* signal.

Point 3.4.2

Figure 3B - Consider adding an additional circle or arrow indicating missing expression in the Schwann cell on the *col18a1a* panel. Alternatively, an overlaid multichannel image would nicely visualize the point made in the text.

Response:

I have modified the panels in Fig. 3B-D to outline the missing expression of *col18a1a* in the Schwann cells adjacent to denervated neuromasts. I have also provided an overlaid multichannel image in the right panel of the figure to emphasize the disappearance and return of the *col18a1a* signal at the different stages of regeneration.

Point 3.4.3

Figure 3D - A stacked bar chart or a bar chart like 3F might be a more intuitive representation of this data. Are there Schwann cells that are negative for *col18a1a*? How do these numbers and expression levels look in intact versus injured animals?

Response:

I have modified the panel to include a stacked bar chart that also details the number of *col18a1a*⁺/*sox10*⁺ Schwann cells after the neuromast has been denervated and reinnervated (Fig. 3F) and also have included the mean and standard deviation of the *col18a1a*⁺ Schwann cell counts in the main text (Line 212, 223, 225). Before lesioning the nerve, I observed a small minority of innervated neuromasts with no adjacent *col18a1a* Schwann cells (3 out of 46 neuromasts). This number rose to 23 out of 26 denervated neuromasts with no adjacent *col18a1a* Schwann cells. The number of reinnervated neuromasts with no adjacent *col18a1a* again drops to only 5 out of 26 neuromasts.

Intriguingly, I counted slightly more *col18a1a*⁺ Schwann cells adjacent to the reinnervated neuromasts (1.50 ± 1.10 , mean \pm SD) compared to before nerve lesion (1.02 ± 0.45 , mean \pm SD). The Schwann cells that are *col18a1a*⁺ in reinnervated neuromasts are still directly adjacent to the location of axonal defasciculation and also lining the pathway of defasciculated axons. While the reason for this increase remains unknown, the changing expression pattern of *col18a1a* following nerve lesion further suggests that the gene is playing a role in patterning nerve regeneration.

Second decision letter

MS ID#: dev.205054R1

MS TITLE: Axonal defasciculation is restricted to specific branching points during regeneration of the lateral line nerve in zebrafish

AUTHORS: Rohan S. Roy and Albert James Hudspeth

Dear Dr Roy,

I have now received all the referees reports on the above manuscript, and have reached a decision. The referees' comments are appended below.

The overall evaluation is positive and we would like to publish a revised manuscript in Development. Reviewer 1 has some comments they would like addressed that are likely to improve the manuscript. Please address these in a response letter and with final revisions to your manuscript. If you do not agree with any of their criticisms or suggestions explain clearly why this is so.

Reviewer 1

Advance summary and potential significance to field

I am satisfied with the changes that the author made to this manuscript, but suggest attention to some minor points that will improve clarity. See below.

Comments for the author

1. Please include individual data points in box and whisker plots: 1D
2. The figure legend for figure 1 includes n=14 zebrafish for 1 μ M AG1478, but the line below lists n=16 zebrafish and includes the 1 μ M treatment in the list of conditions. Please clarify.
3. I cannot see the agrn mRNA signal in the merge in figure S5A.
4. Line 143-144, the author states "We asked whether blocking ErbB signaling from nerve Schwann cells at 4 dpf...", yet as this was a drug application, the manipulation was not specific to any particular cell type, and therefore the drug could be acting on other cells.
4. I am not certain how to interpret plots S6D & E. What does the count on the y axis refer to? Is this a line scan?

Reviewer 2*Advance summary and potential significance to field*

The authors have addressed all of my comments and I find the revised manuscript suitable for publication.

Reviewer 3*Advance summary and potential significance to field*

This study characterizes the role of neuron-extrinsic factors in regulating axon pathfinding and neuromast reinnervation during lateral line nerve regeneration in zebrafish. Expression analysis revealed an intriguing expression pattern for col18a1a near gaps in the ECM where defasciculation occurs. Mutant assays showed a requirement for col18a1a in axon growth patterning, despite normal regeneration and defasciculation. Strengths of this study include the significance of implicating neuron-extrinsic factors (col18a1a and Schwann cells) in pLL regeneration and defasciculation, thorough characterization of axon defasciculation around ECM gaps and a clever experiment showing preferred defasciculation by natural ECM gaps compared to experimentally created gaps.

Comments for the author

The authors did a great job revising the manuscript with new experiments, analyses and text edits. This reviewer has no further comments or suggestions.

Second revisionAuthor response to reviewers' comments

Dear Dr. Poss,

I am delighted to hear that you would like to publish a revised manuscript in *Development*. I have addressed the minor concerns of Reviewer 1 below. I believe the newly revised manuscript further improves the clarity of the results. All changes in the main text and figures have been highlighted in yellow. Changes made to the supplemental figures are explicitly written in this response letter. I am thankful for the thoughtful comments from all the reviewers throughout this process.

Reviewer 1

I am satisfied with the changes that the author made to this manuscript, but suggest attention to some minor points that will improve clarity. See below.

Point 1.1

1. Please include individual data points in box and whisker plots: 1D

Response:

I have added the individual data points to the box and whisker plot in **Fig. 1D**.

Point 1.2

2. The figure legend for figure 1 includes n=14 zebrafish for 1 uM AG1478, but the line below lists n=16 zebrafish and includes the 1 uM treatment in the list of conditions. Please clarify.

Response:

I apologize for the typo. All of the conditions had an $n = 16$ except for the $2 \mu\text{M}$ treatment condition, which had an $n = 14$. The individual data points are now displayed for each condition on the box and whisker plot (**Fig. 1D**) and the legend has been corrected (**Line 957**).

Point 1.3

3. *I cannot see the agrn mRNA signal in the merge in figure S5A.*

Response:

The colors in the merged image in **Fig. S5A** became washed out during PDF conversion. I have remade the combined supplemental image file to maintain proper color saturation.

Point 1.4

4. *Line 143-144, the author states "We asked whether blocking ErbB signaling from nerve Schwann cells at 4 dpf...", yet as this was a drug application, the manipulation was not specific to any particular cell type, and therefore the drug could be acting on other cells.*

Response:

I agree that AG1478 could be acting on other cells and have removed the phrase 'from nerve Schwann cells' from the sentence (**Line 143**).

Point 1.5

4. *I am not certain how to interpret plots S6D & E. What does the count on the y axis refer to? Is this a line scan?*

Response:

The plots in **Fig S6D** and **E** are showing a histogram of pixel intensity values in the ROIs defined in **B** and **C**. The two histograms of pixel values from the *sox10* color channel and the *col18a1a* color channel are overlaid. This is not a line scan; it is a histogram. The *y-axis* represents the number of pixels with the given intensity value denoted by the *x-axis*. I have modified the text of the legend to clarify this: "A histogram of the pixel intensities associated with *col18a1a* and *sox10* probe signals in the ROIs defined in (B) and (C) around Schwann cell nuclei."

Third decision letter

MS ID#: dev.205054R2

MS TITLE: Axonal defasciculation is restricted to specific branching points during regeneration of the lateral line nerve in zebrafish

AUTHORS: Rohan S. Roy and Albert James Hudspeth

Dear Dr Roy,

I am happy to tell you that your manuscript has been accepted for publication in Development, pending our standard publication integrity checks.